# Should VLMs be Pre-trained with Image Data?

**Sedrick Keh**[*1] **Jean Mercat**[*1] **Samir Yitzhak Gadre**[1] **Kushal Arora**[1]
**Igor Vasiljevic**[1] **Benjamin Burchfiel**[1] **Shuran Song**[2] **Russ Tedrake**[1,3]
**Thomas Kollar**[2] **Ludwig Schmidt**[2] **Achal Dave**[1]

[1]Toyota Research Institute, [2]Stanford, [3]MIT,
{sedrick.keh,jean.mercat}@tri.global

## ABSTRACT

Pre-trained LLMs that are further trained with image data perform well on vision-language tasks. While adding images during a second training phase effectively unlocks this capability, it is unclear how much of a gain or loss this two-step pipeline gives over VLMs which integrate images earlier into the training process. To investigate this, we train models spanning various datasets, scales, image-text ratios, and amount of pre-training done before introducing vision tokens. We then fine-tune these models and evaluate their downstream performance on a suite of vision-language and text-only tasks. We find that *pre-training* with a mixture of image and text data allows models to perform better on vision-language tasks while maintaining strong performance on text-only evaluations. On an average of 6 diverse tasks, we find that for a 1B model, introducing visual tokens 80% of the way through pre-training results in a 2% average improvement over introducing visual tokens to a fully pre-trained model.

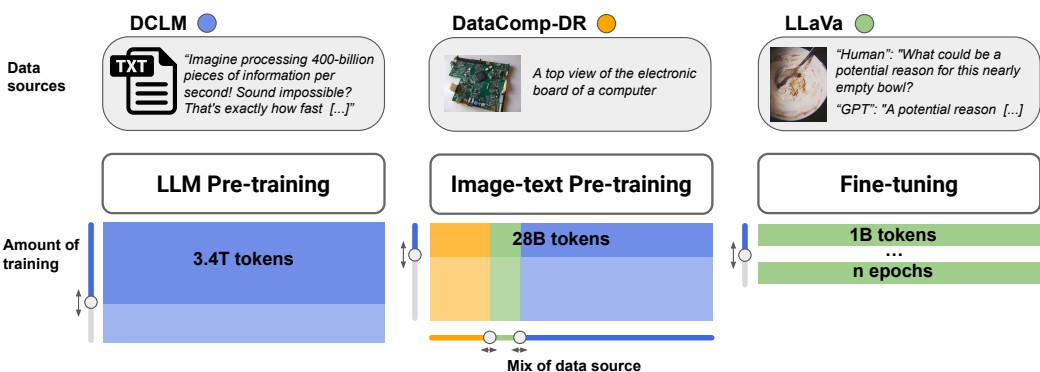

Figure 1: An overview of our VLM pre-training data recipe. We investigate data mixes and design choices for text-only pre-training, image-text pre-training, and fine-tuning. Note that while we depict "LLM Pre-training" and "Image-text Pre-training" as two separate steps in this diagram, in practice, we continuously transition from the first stage to the second.

## 1 INTRODUCTION

Numerous work have extensively studied how to train a VLM from a pre-trained LLM (Liu et al., 2024; Alayrac et al., 2022; Bavishi et al., 2023; Li et al., 2024a; Karamcheti et al., 2024; Tong et al., 2024; Beyer et al., 2024; Laurençon et al., 2024; Lu et al., 2024; Wu et al., 2024a;b; Chen et al., 2025). Most existing VLMs follow roughly the same high-level recipe: 1) Start with a pre-trained LLM, 2) continue training intermediate layers on image-text tokens (e.g., Flamingo, Pali-Gemma, Idefics, DeepSeek-VL) or align an image projection layer (e.g., LLaVA, Cambrian, Prismatic), and 3) fine-tune the model on multi-task instructions or chat templates. Notably, training with images included in the initial pre-training stage (i.e., step 1) is largely undocumented in these papers, even though a number of large models (with undocumented training procedures) are known to be natively multimodal (e.g., Pixtral, Gemini, Fuyu).

In this paper, we investigate the significance of each step in the pipeline for downstream performance on vision-language and text benchmarks. Additionally, we examine whether increasing the amount of image data in one step can reduce the data requirements in other steps. To understand when image data should be introduced to VLM training, we train a suite of 300 models over various numbers of parameters, varying the amount of text-only pre-training data, as well as the amount, type, and ratio of image pre-training data (Figure 1).

Our experiments suggest several key findings:

First, incorporating image data during pre-training generally helps, especially *after* a model has seen many text tokens. However, the timing and method of introducing visual information are also important. Specifically, we find that it is beneficial to add image data before the LLM is fully pretrained, during the cooldown phase. Our strategy to interrupt the text pre-training at 80% completion and add image-text data outperforms the popular alternative of fully training the LLM, then re-warming and cooling the model with image data (Section 3.1).

Second, the fraction of visual data introduced during cooldown is another key parameter for strong performance across domains. At the 1B parameter regime, our experiments reveal that 10% to 20% of tokens should be visual. Going above or below this ratio results in worse downstream performance. However, this fraction appears to be a function of scale. At the smaller 79M parameter regime, larger fractions of visual data are preferred (Sections 3.2 and 3.3).

Third, the timing of *when* to introduce instruction fine-tuning image tokens is crucial for downstream performance, both for pure text and vision-language tasks. We observe that mixing together instruction fine-tuning during the image-text pre-training process actively hurts the model (Section 3.4). Meanwhile, adding instruction tokens during fine-tuning improves vision-language task performance up to 4 epochs at the 1B parameter scale, at the cost of slightly hurting performance on text-only tasks (Section 3.5).

## 2 EXPERIMENTAL SETUP

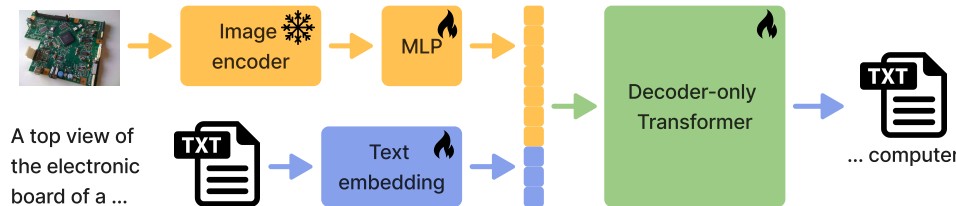

Figure 2: The commonly used framework we apply to add vision capabilities to a transformer model.

Our experimental setup is designed to study the impact of image-text data on the model's downstream results on both text and vision tasks. To this end, we implement a commonly used model architecture (Liu et al., 2023; Tong et al., 2024; McKinzie et al., 2024; Beyer et al., 2024; Bai et al., 2023; Li et al., 2023a; Laurençon et al., 2024) consisting of a pre-trained image encoder, a projection block, and a decoder-only transformer. Unless stated otherwise, the image encoder is SigLIP 400M (Zhai et al., 2023), the projection block is a two-layer MLP, and the transformer is the 1.4B parameter model described in Gadre et al. (2024b). Our training procedures (Section 2.1) are applied to the model prior to its downstream evaluation (Section 2.2).

### 2.1 TRAINING PROCEDURE

We train all our models with a three-step process illustrated in Figure 1:

1. *Partial* text-only pre-training
2. Image-text pairs mixed with text-only *continued* pre-training
3. Image-text pairs multi-task fine-tuning

We can view the first and second steps as a single pre-training stage, where we introduce images *before* the model has finished pre-training, as opposed to previous methods where images are introduced

Table 1: The conventional approach to VLM training first fully trains a language model on text-only data, and adds images in a second or third stage of training. In this work, we instead introduce image data earlier in pre-training. To do this efficiently, we resume a language model during the course of its pre-training at various stages (e.g., 20% of the way through training) and introduce images midway through training.

| Model | Text-Only Pre-training | Image-Text Pre-training | Multitask Fine-Tuning |
|---|---|---|---|
| **BLIP3** (Xue et al., 2024) | Fully pre-trained (Phi3-mini) | Re-warmup Caption and interleaved text-image data; no pure text | Re-warmup |
| **Flamingo** (Alayrac et al., 2022) | Fully pre-trained (closed model) | Re-warmup Caption and interleaved text-image data; no pure text | (Skipped) |
| **IDEFICS** (Laurençon et al., 2024) | Fully pre-trained (Mistral-7B-v0.1) | Re-warmup Interleaved text-image data; no pure text | Re-warmup |
| **MM1** (McKinzie et al., 2024) | Fully pre-trained (closed model) | Re-warmup Various image-text ratios (100::0, 91::9, 86::14, 66::33) | Re-warmup |
| **DeepSeek-VL / DeepSeek-VL2** (Lu et al., 2024; Wu et al., 2024b) | Fully pre-trained (DeepSeek) | Two-stage (re-warmup then re-warmup) Multitask; 30::70 image-text ratio | Re-warmup |
| **Qwen-VL** (Bai et al., 2023) | Fully pre-trained (Qwen) | Two-stage (re-warmup then re-warmup) Caption then multitask; no pure text | Re-warmup |
| **PaliGemma** (Beyer et al., 2024) | Fully pre-trained (Gemma) | Two-stage (re-warmup then continue LR schedule) Multitask; no pure text | Continued LR schedule |
| **Janus / Janus Pro** (Wu et al., 2024a; Chen et al., 2025) | Fully pre-trained (DeepSeek-LLM) | Two-stage (re-warmup then re-warmup) Multitask; no pure text, followed by 30::70 image-text ratio | No warmup |
| **Prismatic** (Karamcheti et al., 2024) | Fully pre-trained (LLaMA) | No image-text pre-training. | Re-warmup |
| **Ours** | **Partially** pre-trained (OpenLM) | Continued LR schedule Various image-text ratios | Re-warmup |

to a model which had already been pre-trained. In practice, this is implemented by resuming from the same learning rate schedule (Section 2.1.2). A more detailed comparison overview of different models can be found in Table 1.

### 2.1.1 TEXT-ONLY PRE-TRAINING

During the initial pre-training process, only the LLM backbone is trained with the training recipe from Gadre et al. (2024b). Rather than re-training from scratch, we instead leverage existing open checkpoints from DCLM-1B (Li et al., 2024b) [1]. This is a 1.4B parameter model trained on text tokens from DCLM-Baseline mixed with StarCoder (Li et al., 2023b) and ProofPile (Azerbayev et al., 2024), for a total of 4.3T tokens, and it outperforms the recent LLaMA-3.2 1B model [2] on several commonly used text benchmarks.

The pre-training learning rate schedule is warmup-cosine with a peak learning rate of $10^{-2}$ and a final learning rate of $10^{-5}$. In addition to using the final model, we also use checkpoints from 20%, 40%, 60%, and 80% through training, which correspond to 860B, 1.72T, 2.58T, and 3.44T training tokens respectively. Since these checkpoints are taken from the middle of training, their learning rates are higher than the typical last learning rates of most released model weights, allowing us to effectively resume pre-training (Figure 3).

Additional hyperparameters for the text-only pre-training stage can be found in Appendix A.

### 2.1.2 IMAGE-TEXT PRE-TRAINING

We take checkpoints from the text-only pre-trained model and resume the cosine learning rate schedule with a cooldown phase, as illustrated in Figure 3.

For our experiments, we perform multi-modal training using both caption data and task-related instruction data. As opposed to other VLMs like MM1 (McKinzie et al., 2024), BLIP3 (Xue et al., 2024), and IDEFICS (Laurençon et al., 2024) which use interleaved data in addition to caption data, we limit our setting to either text only or text + a single image. Our minimal setting allows us to control as many factors as possible to isolate the most fundamental factors to train a performant multi-modal model.

---

[1] `https://huggingface.co/TRI-ML/DCLM-1B`
[2] `https://huggingface.co/meta-llama/Llama-3.2-1B`

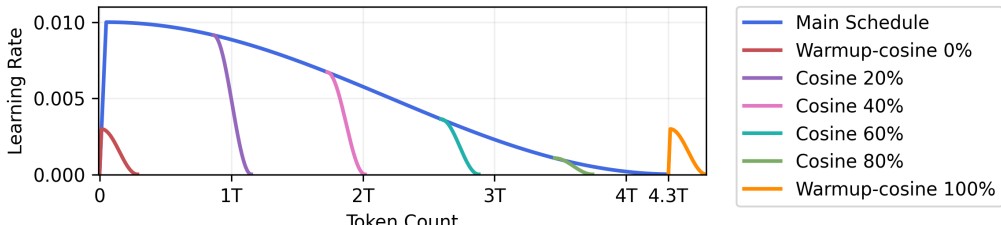

Figure 3: Representation of the different learning rate schedules used for our experiments. 'Main schedule' corresponds to the learning rate for the initial, text-only pretraining. Other colored schedules are the ones used for image-text training and extend over 28B tokens each. They have been upscaled and appear as extending over 280B tokens for readability.

There are a number of design choices involved in this stage. We give an overview of these design choices below:

**Text dataset selection:** During the image-text pre-training stage, we train with the DCLM-baseline dataset, mixed with StarCoder and MathPile. This is selected to match the dataset used during the text-only pre-training step to ensure continuity from the LLM checkpoints that we resume.

**Caption dataset selection:** We train with the DataCompDR-1B caption dataset (Vasu et al., 2024), which is an enhancement over DataComp-1B (Gadre et al., 2024a) by regenerating higher quality captions. To arrive at this selection, we ablated over several caption datasets (Appendix J.1).

**Image-text ratio:** The effect of mixing text and images has not been thoroughly documented and is one of the key questions we explore in this paper. McKinzie et al. (2024) show that adding some text tokens helps with few-shot and text-only performance, though they show only up to 33% text and operate at slightly different settings (fully pre-trained vs partially pre-trained model). DeepSeek-VL (Lu et al., 2024) uses a fixed 70% text ratio but does not discuss other ratios. We conduct a thorough investigation on the image-text ratio in Sections 3.3 and 3.2.

**Amount of training**: We train all our models with a token multiplier of 20, following approximate Chinchilla optimal scaling (Hoffmann et al., 2022). For a 1.4B model, this equates to around 28B tokens. From our experiments at smaller scale, we observe that trends generally hold across higher token multipliers (Appendix G).

**Image encoder selection:** We found the SigLIP encoder to work well for our setting, which confirms findings from Beyer et al. (2024). The DINO+SigLIP combination from Karamcheti et al. (2024) gave slightly better results on some benchmarks; for simplicity, however, we use SigLIP for most of our experiments. See Appendix J.2 for full ablations.

**Frozen encoder weights:** The vision encoder can be kept frozen (Karamcheti et al., 2024) (i.e., its parameters are not trained) or trained end-to-end with the language model (Tong et al., 2024; Beyer et al., 2024). In our small-scale ablations, we found it more effective to freeze the image encoder weights in both the pre-training and fine-tuning stages (Appendix I), so we freeze the weights for the rest of this section.

Each image is encoded to 729 tokens, and we train all models for a sequence length of 1024. We provide additional implementation details in Appendix C.

### 2.1.3 INSTRUCTION FINE-TUNING

We fine-tune our pre-trained models with the LLaVA dataset (Liu et al., 2024) using the Prismatic framework (Karamcheti et al., 2024). For each model trained in the previous step, we fine-tune for $\{1, 2, 3, 4\}$ epochs. We use a cosine learning rate schedule with warmup. To account for different epoch numbers having different learning rate schedules, we treated each epoch number as its own separate run. A full list of hyperparameters can be found in Appendix D.

## 2.2 EVALUATION

We want our VLMs to perform well in vision-language tasks while still retaining their performance in text-only tasks. As such, we conduct evaluations over a suite of diverse tasks, both image-based and text-based. In our early experiments, we trained a 79M parameter model with text and used it to search the training space before scaling to the 1B experiment presented in this work. We only keep the downstream text tasks where a 79M parameter model performs better than random chance. We test our model without any in-context learning, thus all results reported are 0-shot.

**Vision-language tasks:** We evaluate on a subset of vision-language tasks used by Karamcheti et al. (2024)[3]. These tasks test general visual reasoning, spatial reasoning, ability to read text, object localisation and hallucination. For a full set of vision-language evaluation tasks, see Appendix E.

In addition to reporting the scores on individual tasks, we also report an aggregate metric. All our results are accuracy scores (between $[0, 1]$). To aggregate them, we subtract the random baseline accuracy, then average the result over all tasks. We call this aggregate metric the *stable score*. Its absolute value can be interpreted as the percentage points of advantage over random guessing. Its trend can be understood as average percentage points on the suite of benchmarks.

**Text tasks:** We evaluate on a suite of tasks taken from Gadre et al. (2024b) and conduct our evaluations with Eleuther's LM Harness (Gao et al., 2024). These tasks test for general and specific knowledge, reasoning, and commonsense. We also compute the *stable score* as previously described. For a full list of our text evaluation tasks, see Appendix E.

## 3 RESULTS

This section presents a series of experiments exploring different aspects of multi-modal training. We aim to find important choices that affect the downstream performance of the model on a set of diverse visual tasks and text-only question answering.

All experiments below are conducted on the 1.4 billion parameter regime. We make use of various checkpoints of the DCLM-1B model along its text-only pre-training. This is explored in more detail in Section 3.1 below.

All experiments use in different proportions three dataset sources. (1) A text-only dataset, DCLM-baseline (Li et al., 2024b), which was used to pre-train the 1B language model. (2) An image caption dataset, DataCompDR-1B (Vasu et al., 2024), and (3) An image instruction tuning dataset, mix of openly available data described in *A.1/Multimodal Instruct Tuning* (Karamcheti et al., 2024).

Each section below discusses a different set of experiments with a given setup summarized in a green box labeled *Setup*. The setup describes which checkpoint of the text-only pre-trained model is used, the mix of text and image captioning data used for image-text pre-training, and the number of fine-tuning epochs performed on the image instruction tuning dataset. Within each description, we use $x$ to denote the variable that we are changing for that particular experiment, while keeping the rest of the variables constant.

For each of the sections below, there will be two plots, each with two vertical axes, representing our experiment results on vision-language and text tasks. Each point on a plot represents a 1B model initialized from one of the text-only pre-trained checkpoints, pre-trained with text and image captions, and fine-tuned with image instruction tuning data. The plot on the left represents the stable score, an aggregate of performance results, across the vision benchmark on one hand and text benchmarks in the other. The plot on the right represents one of each set of benchmark, the VQA-v2 (Goyal et al., 2017) results and the ARC-easy (Clark et al., 2018) results. These might be easier to interpret but do not convey the full extent of the evaluation. The full list of benchmarks as well as detailed graphs for the vision performance can be found in Appendix F.

---

[3]https://github.com/TRI-ML/vlm-evaluation/

## 3.1 THE IMPACT OF AMOUNT OF TEXT-ONLY PRE-TRAINING

**Setup**
- $x\%$ checkpoints of 1B model
- Image-text pre-trained for 28B tokens:
  - 90% Text-only
  - 10% Image captions
- Fine-tuning 4 epochs

For this experiment, we use different checkpoints along the text-only pre-training. The initial text model was trained with a warmup-cosine learning rate schedule. All continued image-text pre-training runs follow the learning rate schedules represented in Figure 3. We continue training starting from the last text-only learning rate value, if not too low, and continue training for 28B. For 0% and 100% the learning rate would be too low, thus we adopt a linear warmup-cosine decay with a maximum learning rate of $3 \times 10^{-3}$. See Figure 3.

Figure 4 below shows the progression of the model performance as a function of the text pre-training completion of the initial checkpoint. We see that up to 80%, a larger amount of text-only pre-training improves performance for both text and vision tasks, suggesting that it's helpful to start with a stronger model which has been trained for a greater number of text tokens. However, at 100% the learning rate schedule is different and could affect the results. Many similar works all initialize their model to a 100% text-only pre-trained model and all re-warmup the learning rate during the image-text pre-training phase. From our experiments, we observe a decrease in performance at 100%, suggesting that continued training is preferable to re-training a 100% fully pre-trained text model.

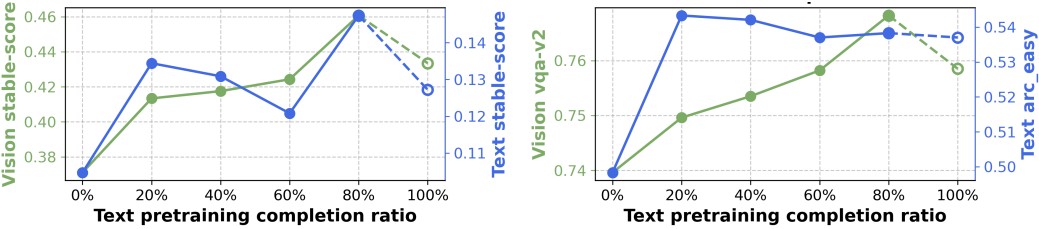

Figure 4: **Varying the length of text-only pre-training.** We analyze the impact of adding image data after varying amounts of text-only pre-training, showing results on vision benchmarks (green) and text benchmarks (blue). On the left, we show results across a suite of vision and text benchmarks; on the right, we plot two common benchmarks, VQA-v2 and ARC-easy. Introducing images at around 80% of the way through training performs best, maintaining high vision and text task performance. Note: The points at 100% are marked with hollow circles to highlight that they are trained with a different learning rate schedule, as shown in Figure 3

> Longer text pre-training improves performance but, when adding image input, continued training gives better performance than re-warming up.

## 3.2 THE IMPACT OF ADDING IMAGES BEFORE THE END OF PRE-TRAINING

**Setup**
- 80% pre-trained 1B model
- Image-text pre-trained for 28B tokens:
  - $1 - x\%$ text
  - $x\%$ image captions
- Fine-tuning 4 epochs

Building on the results from Figure 4, we take the 80% checkpoint and vary the image-text ratio we add during pre-training. Figure 5 shows the evolution of the model performance as we pre-train (from a text-only pre-trained model) with more image data. We keep the total number of pre-training tokens constant at 28B, adjusting only the image-text ratio.

From Figure 5, we see that as more image data is added to the mix, better downstream performance is obtained on vision tasks. However, text-only downstream performance decreases. When only image data is used for image pre-training, both text and vision task performance drop significantly. We observe that from 0% images (i.e. train on 100% text) to 1% images, there is a slight drop in vision stable score performance. We believe this can possibly be attributed to the model being confused at the addition of this new task, and that it requires

some minimum number of tokens in order to truly learn the image representations. Interestingly, performance on text-only tasks gets slightly better when some image data is added to the pre-training mix.

> Introducing image data at a late stage during pre-training improves vision performance while maintaining text performance.

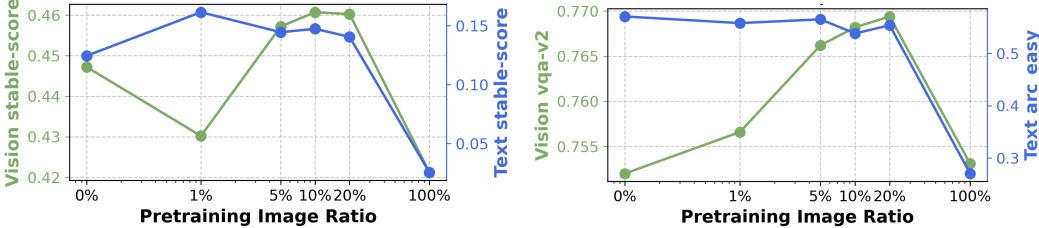

Figure 5: **Varying the ratio of image to text data, after some text-only pretraining.** We analyze the impact of the ratio of image to text data in pre-training, after the model has seen text-only data for most of pre-training (80%). Unlike when training from scratch (Figure 6), we find that adding vision data significantly helps vision performance, while maintaining high text accuracy.

### 3.3 THE IMPACT OF ADDING IMAGES WHEN TRAINING FROM SCRATCH

**Setup**
- 0% randomly initialized 1B model
- Image-text pre-trained for 28B tokens:
  - $1 - x\%$ text
  - $x\%$ image captions
- Fine-tuning 4 epochs

This section mirrors Section 3.2, with the only change being starting the image-text pre-training from scratch as opposed to from an 80% checkpoint. Figure 6 shows the evolution of the model performance as we pre-train the model from scratch with more image data.

We observe slighly different trends for models trained from scratch (Figure 6) as compared to the models trained from the 80% checkpoint (Figure 5). Here, as more image data are added to the mix, we obtain worse downstream performance on both text and vision tasks. These models were trained for 28B tokens each and would likely benefit from more training; it is possible that at larger scales, these trends will more closely resemble those in Figure 5. Interestingly, just like in Figure 5, adding 1% image data strongly degrades the ability of the model to be fine-tuned for vision tasks.

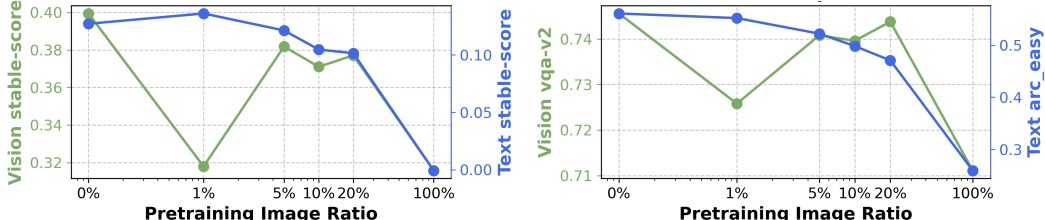

Figure 6: **Varying the ratio of image to text data, when training from scratch.** We analyze the impact of the image-text ratio in pre-training from scratch without any language-only pre-training. Perhaps surprisingly, when training from scratch, adding vision data consistently hurts both vision and text performance, suggesting a period of language-only training early on is important for VLMs.

> Introducing image data when *training from scratch* at small scales does not improve downstream vision performance.

## 3.4 THE IMPACT OF INSTRUCTION TUNING DATA IN PRE-TRAINING

**Setup**
- 80% pre-trained 1B model
- Image-text pre-trained for 28B tokens:
  - 90% text
  - $10 - x\%$ image captions
  - $x\%$ image instruction tuning data
- Fine-tuning for 4 epochs

We train a 1B model with a mix of 10% image-text and 90% text-only data and vary the source of the image-text data from purely image captions to purely instruction tuning data. Figure 7 shows the evolution of the model performance as we increase the proportion of image instruction tuning in the mix of image-text data. We observe that the best performance on vision-language tasks occurs when there are no instruction tuning data included in the pre-training mix, although adding instruction-tuning data seems to positively affect scores on text-only tasks. From a vision-langauge standpoint, we conclude that image instruction tuning data should not be added to the pre-training mix and only be used in a separate fine-tuning stage.

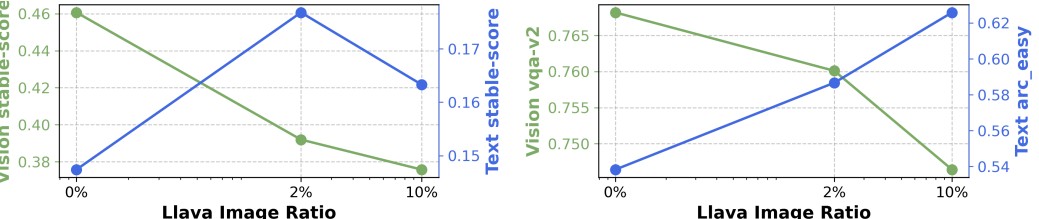

Figure 7: **Varying the proportion of instruction tuning data in the image mix.** Is including instruction tuning data during pre-training is helpful for VLMs? Surprisingly, we find that adding this data to pre-training *harms* performance. We hypothesize that this may be due to overfitting, or because mixing instruction tuning data with image-caption pairs degrades learning at this scale.

> The presence of image instruction tuning data in pre-training harms downstream evaluations.

## 3.5 THE IMPACT OF FINE-TUNING ON VISION AND TEXT PERFORMANCE

**Setup**
- 80% pre-trained 1B model
- Image-text pre-trained for 28B tokens:
  - 90% text
  - 10% image captions
- Fine-tuning for $x$ epochs

Figure 8 shows the evolution of the model 1B performance as it is fine-tuned for 1 to 6 epochs. Looking at the stable score, we observe that up to 4 epochs, downstream vision task performance improves and the downstream text performance degrades. Beyond 4 epochs of fine-tuning, both vision and text performance degrade, likely due to overfitting.

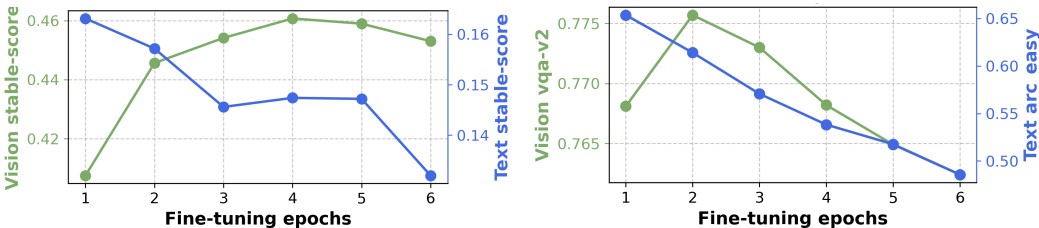

Figure 8: **Varying the number of fine-tuning epochs.** We find that fine-tuning for 2-4 epochs after pre-training performs best for vision tasks, with 2 epochs being a sweet spot for maintaining text performance while achieving high vision performance.

> Fine-tuning for image instruction improves vision-language tasks even beyond one epoch, up to a point, but degrades text-only task performance.

# 4    RELATED WORK

The related work covers three subjects, first, two-stage vision language models (VLMs), which rely on pre-training with text-only data, then recent VLMs which introduce vision data earlier in pre-training, and finally, multimodal datasets for VLM training.

**Two-stage VLMs.**    The standard approach for training vision language models involves first training a large language model on text-only data, and then incorporating vision data in a second fine-tuning phase. This strategy is employed effectively by models including Liu et al. (2024; 2023) (LLaVA and variants), Alayrac et al. (2022) (Flamingo), Li et al. (2023a); Xue et al. (2024) (BLIP-2 and BLIP-3), Laurençon et al. (2024), Laurençon et al. (2023) (IDEFICS and Obelics), Chen et al. (2025) (Janus and Janus-Pro) among others. McKinzie et al. (2024) explore design choices for training VLMs complementary to the ones we assess, namely the vision-language connector choice, the pretraining strategy for the image encoder, and the image resolution, while using a two-stage training strategy with a fixed, pre-trained language model.

**Pre-training with visual data.**    Bavishi et al. (2023) (Fuyu-8B) is a notable exception which trains a vision-language model from scratch, but the training strategy is not publicly released. Team (2024) introduces a series of early-fusion VLMs, which, similar to our work, introduce image data during pre-training, but choose a specific, handcrafted strategy for training. For multi-modal pre-training, Aghajanyan et al. (2023) find scaling laws for pre-training from text, code, image-text, image and speech data in terms of unimodal and bi-modal combinations. They investigate training dynamics for various mixtures of pre-training data and model sizes in terms of perplexity. Their study is related to ours in the sense of pre-training with mixtures of data, but it differs in that it does not involve any downstream evaluations. We focus specifically on the setting of VLM training, and thoroughly investigate the results on standard benchmarks.

**Datasets**    Collecting image-text pre-training and instruction data is challenging, due to the low quality of image-text pairs on the internet. Most datasets use a combination of curation and re-captioning to address this, including Desai et al. (2021) (RedCaps), or Thomee et al. (2015) (YFCC-100M). In this work, we specifically rely on recent large-scale, high-quality datasets that perform well for training multimodal models: Changpinyo et al. (2021) (ConceptualCaptions-12M) follows the ConceptualCaptions-3M (Sharma et al., 2018) pipeline, curating images and alt-text pairs from the internet, with relaxed filters to increase the dataset size. Gadre et al. (2024a) (DataComp-1B) uses CLIP-based filtering to curate a high-quality set of 1 billion image-text pairs, and Vasu et al. (2024) (DataCompDR-1B) uses a vision-language model to generate synthetic captions for the original DataComp-1B dataset. Li et al. (2024c) (Recap-DataComp-1B) further improves on this by using a LLaMA3-based LLaVA-1.5 model (Liu et al., 2023) to perform the recaptioning. For finetuning, we use the LLaVA-1.5 (Liu et al., 2024; 2023) mix, which consists of high quality, curated image datasets, along with instruction-tuning data generated using a large language model. Recent datasets further explore *interleaved* data, where the same sequence counts several images and text piece that relate to each other. Interleaved datasets include OBELICS (Laurençon et al., 2023), the Cauldron (Laurençon et al., 2024), and MINT-1T (Awadalla et al., 2024). We leave the analysis of this data for VLM pretraining to future work.

# 5    CONCLUSION

In this paper, we address some key gaps and challenge several common assumptions in vision-language model (VLM) pre-training. Specifically, our work questions the traditional practice of separating text and image pre-training phases, demonstrating that a more integrated approach of incorporating image data during pre-training can yield superior downstream results. We recommend future VLM efforts leverage intermediate pre-training checkpoints to incorporate images before completing the pre-training process. In this setting, it's important to carefully manage the image-to-text ratio: unlike the typical approach where a separate image training stage is done with 100% caption images, this integrated setting requires a balanced image-text ratio to avoid performance degradation. We plan to make our code and our testbed of models publicly available, and we hope that our findings will provide a strong empirical foundation for open-source VLM pre-training.

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

## A  TEXT-ONLY PRE-TRAINING HYPERPARAMETERS

We preformed initial exploratory experiments with a 79M parameter model and scale our final experiments to a 1B model with settings from Gadre et al. (2024b) given in Table 2. Our text models are trained with the OpenLM (Gururangan et al., 2023) codebase for next token prediction on the DCLM-baseline dataset (Li et al., 2024b).

Table 2: **The two models and set of hyperparameters used in our experiments.** Models have number of parameters $N$, with number of layers $n_{layers}$, number of attention heads $n_{heads}$, model width $d_{model}$, and width per attention head $d_{head}$. Batch sizes are global and in units of sequences. Each sequence has 2,048 tokens. A100 GPU hours are at $M = 150$. For the 1.4B scale, a batch size of 256 performs slightly better than 512.

| $N$ | $n_{layers}$ | $n_{heads}$ | $d_{model}$ | $d_{head}$ | Warmup | Learning rate | Batch size | Training tokens | A100 hours |
|---|---|---|---|---|---|---|---|---|---|
| 79M | 8 | 4 | 512 | 128 | 400 | $3e$-3 | 512 | 237B | 1.2k |
| 1.4B | 24 | 16 | 2,048 | 128 | 5000 | $1e$-2 | 256 | 4.3T | 106k |

## B  IMAGE-TEXT PRE-TRAINING HYPERPARAMETERS

We add an image encoder, loaded from a pre-trained model (or randomly initialized, in the case of patch projection). We define a 2-layer MLP with GELU activation to project the image patches from the image encoder dimension to a dimension 4 times larger and then down to the LLM input dimension.

Image-test pre-training follows the same procedure as text pre-training except that the image patches are not predicted by the auto-regressive model and are masked out of the auto-regressive cross-entropy loss.

We use the learning rate schedules defined in Figure 3.

## C  IMAGE ENCODING IMPLEMENTATION DETAILS

We encode each image to the vocabulary space of the language model, with each image taking 729 tokens. When dealing with caption data, we always place the image first before the text, and we add a separator token to separate the image and the text. We then mask out the image and only compute loss over the text tokens. However, we do not use a PrefixLM architecture as used in Beyer et al. (2024) and Zhou et al. (2024). We believe this is a promising direction to explore in the future.

Meanwhile, when dealing with LLaVA data, we do a similar process of adding the image before the text and masking the image. This time, we also make sure to mask out any system prompt and human question, only keeping the model responses unmasked. We also add a separator token at the end of every human question and model response.

## D  FINE-TUNING HYPERPARAMETERS

As described in Karamcheti et al. (2024), our image-text instruction tuning dataset is composed of 665K examples. Each sequence that is fed to the model begins with 729 items matching image patch embedding followed by different numbers of embedded text tokens. The model is trained using the Prismatic codebase Karamcheti et al. (2024). Instruction tuning follows the same training strategy as pre-training except that the image patch and question tokens are masked out of the cross-entropy loss.

We use the following non-default hyperparameters:

- Learning rate: $3.10^{-4}$
- Warmup ratio: $0.05$
- Adam optimizer: $\beta_2 : 0.95$
- Finetune epochs: $4$

- Batch size: 256
- Image resize strategy: "letterbox"

## E  EVALUATION BENCHMARK DESCRIPTIONS

**VQA benchmarks:**

- VQAv2 (Goyal et al., 2017): General visual reasoning
- GQA (Hudson & Manning, 2019): Spatial reasoning
- TextVQA (Singh et al., 2019): Text-based reasoning
- POPE (Li et al., 2023c): Yes/No hallucination test

**Object localisation benchmarks:**

- RefCOCO (Kazemzadeh et al., 2014; Yu et al., 2016): Object localisation
- OCID-Ref (Wang et al., 2021): Cluttered object localisation

These are a subset of the tasks considered in Karamcheti et al. (2024). We evaluate and report VizWiz results but leave them out of the average performance computation because its system prompt is out of the training distribution and is often misunderstood by the 1b model. The "unanswerable" answer suggested in the prompt is rarely used by the model, which would often answer reasonable but different responses such as empty responses or "I don't know". The exact match scoring does not reflect that, causing a large variance of the results. With different initial random seeds, most results stay similar, but VizWiz results vary by several percentage points.

We also left out challenge tasks VSR, TallyQA and AI2D because the difficulty of these tasks make them out of scope for small scale models.

**Text benchmarks:**  The following benchmark descriptions are taken from Li et al. (2024b)

- AGI Eval LSAT-EN (Zhong et al., 2023) tests for model knowledge in the legal domain and evaluates analytical reasoning capabilities.
- ARC-easy (Clark et al., 2018) contain four-way multiple choice questions taken from grade 3-9 science exams, where questions in the easy dataset require knowledge of basic science, and the challenge questions require some procedural reasoning.
- BigBench (bench authors, 2023) Conceptual combinations 4-way multiple choice question answering dataset which discriminate combinations of objects and attributes that are appropriate to each other or not.
- BoolQ (Clark et al., 2019) binary question answering dataset where the model is expected to answer questions about relevant passages.
- COPA (Roemmele et al., 2011) causal reasoning questions where the model is given two possible outcomes to a scenario and must use commonsense to select the outcome that is more likely.
- HellaSwag (Zellers et al., 2019) a conversational question answering dataset where the model is given a passage and conversation between two participants and then expected to extract an answer from the passage to a question from one of the participants.
- MathQA (Amini et al., 2019) 5-way multiple choice question answering dataset that evaluates math word problem solving capabilities, built on top of AQuA.
- PIQA (Bisk et al., 2020) binary multiple choice question answering dataset that requires the model to use physical commonsense reasoning to answer correctly.
- PubMedQA (Jin et al., 2019) 3-way multiple choice question answering dataset which evaluates the model's ability to answer biomedical research questions given context from a relevant research article.

# F DETAILED RESULTS

## F.1 THE IMPACT OF TEXT-ONLY PRE-TRAINING

Figure 9 shows the detailed vision benchmarks results and global text benchmark results as the initial text-only pre-trained model is trained to completion. All tasks except VizWiz perform the best with the text backbone trained to 80% of completion.

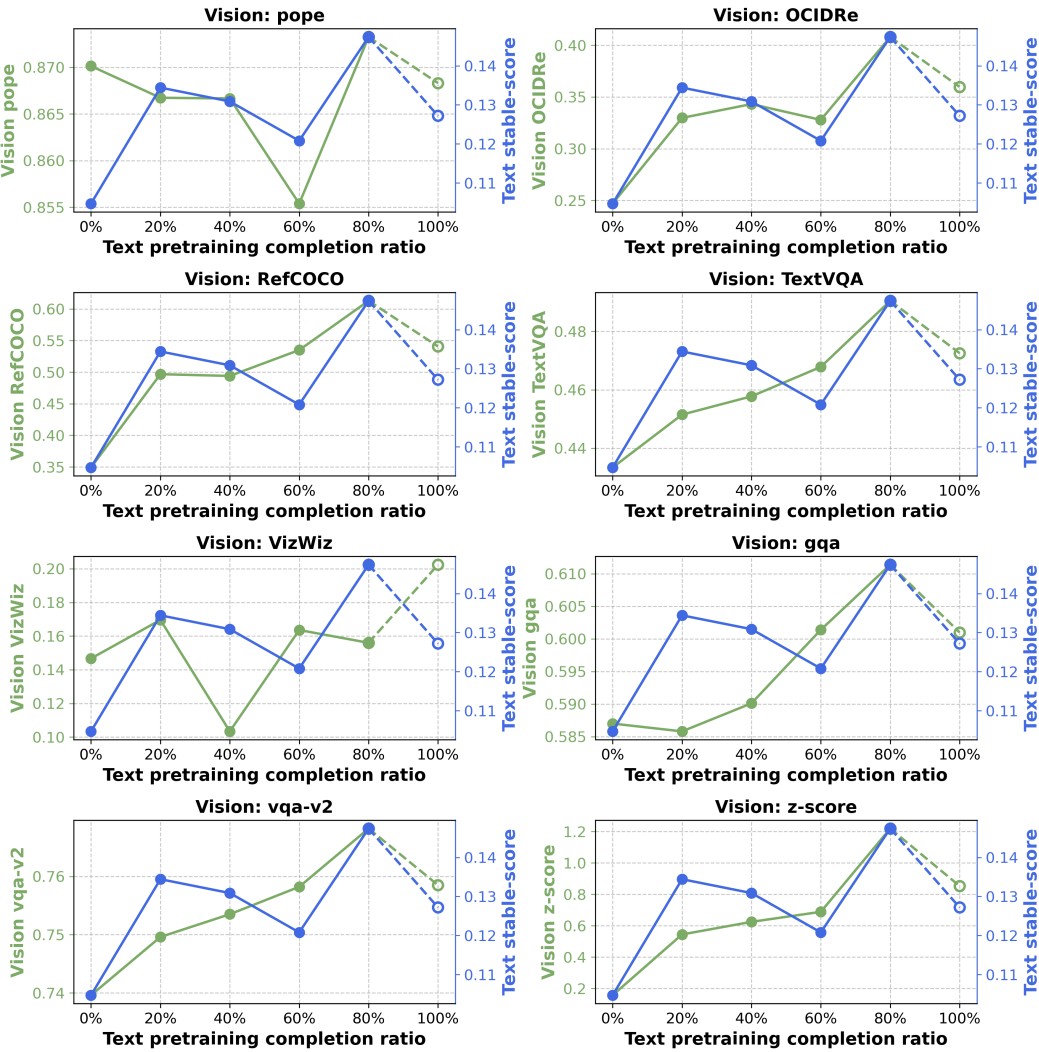

Figure 9: Evolution of the performance of the 1b model on vision benchmarks and text benchmarks as functions of the text-only pre-training completion.

## F.2 THE IMPACT OF ADDING IMAGES BEFORE THE END OF PRE-TRAINING

For some tasks such as POPE and RefCOCO, the best performance is obtained without any image in the second stage. Most other tasks do benefit from image-text pre-training.

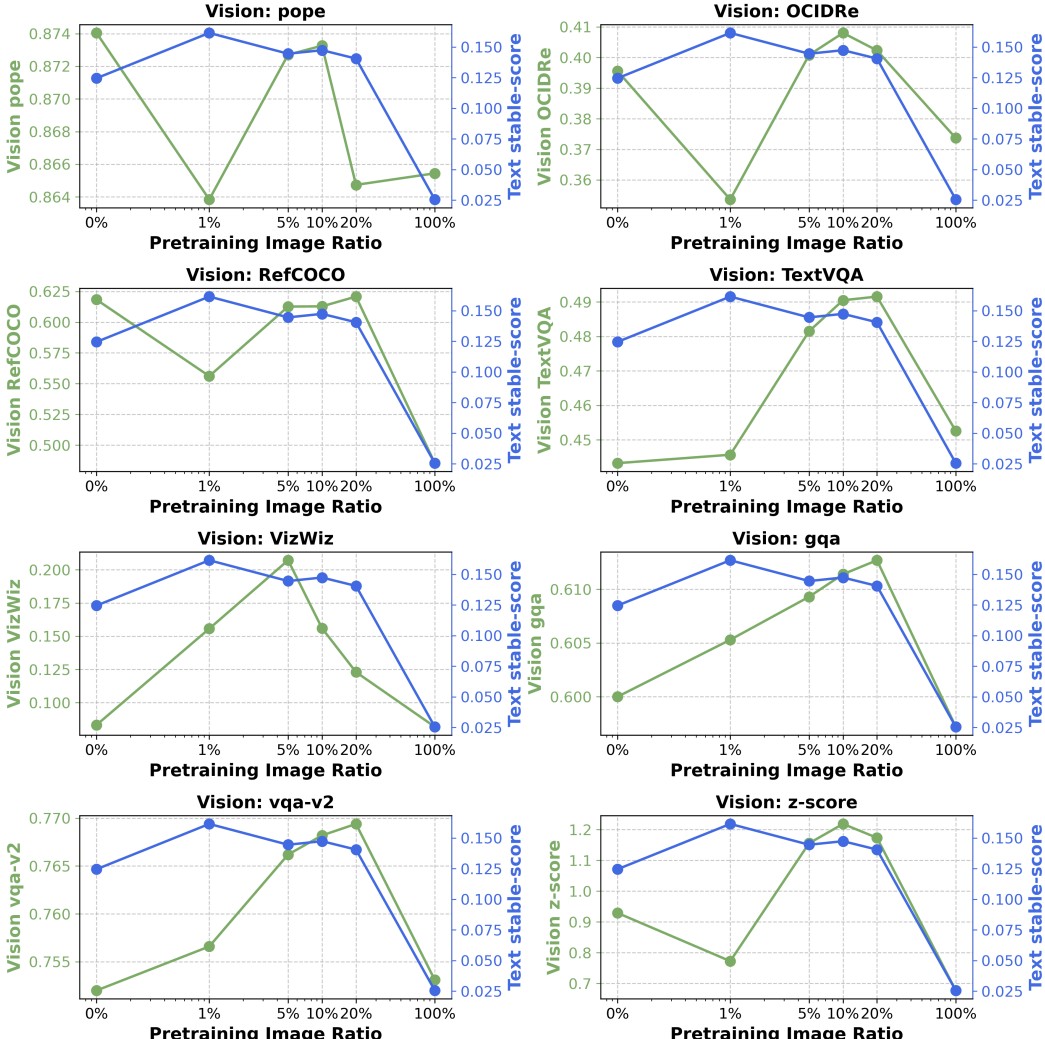

Figure 10: Evolution of the performance of the 1b model on vision benchmarks and text benchmarks as functions of the ratio of image caption data in the image pre-training phase.

F.3    THE IMPACT OF FINETUNING ON VISION AND TEXT PERFORMANCE

The trend of progress over finetuning for individual benchmark is only clear for some benchmarks such as RefCOCO and OCID. For VQA-v2, GQA, TextVQA, and POPE, the best results are not at 4 epochs but less than one percentage point is lost between their maximum performance and their performance at epoch 4.

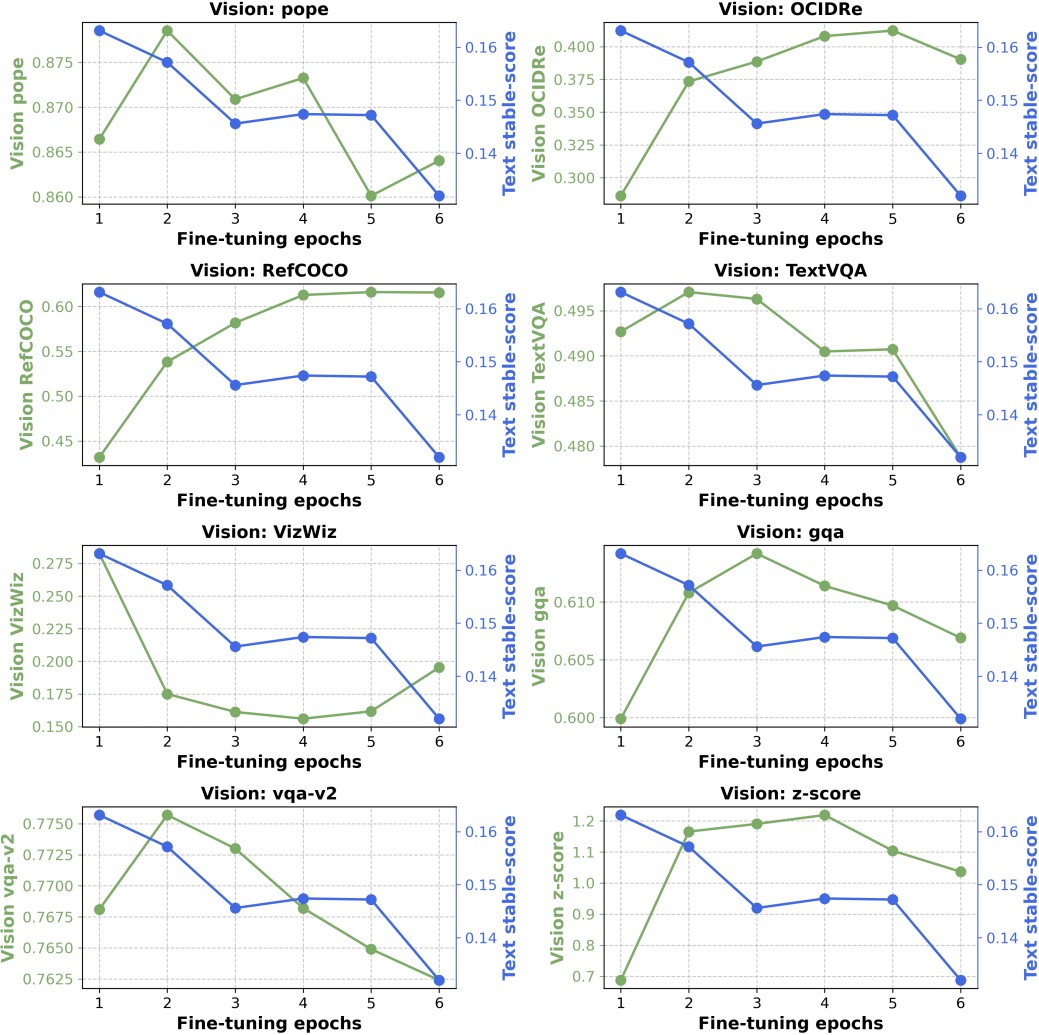

Figure 11: Evolution of the performance of the 1b model on vision benchmarks and text benchmarks as functions of the number of fine-tuning epochs.

## F.4 THE IMPACT OF ADDING IMAGES FROM THE START WHEN TRAINING FROM SCRATCH

Although not by a large margin, in this pre-training from scratch experiments, best results are obtained without image in the pre-training mix.

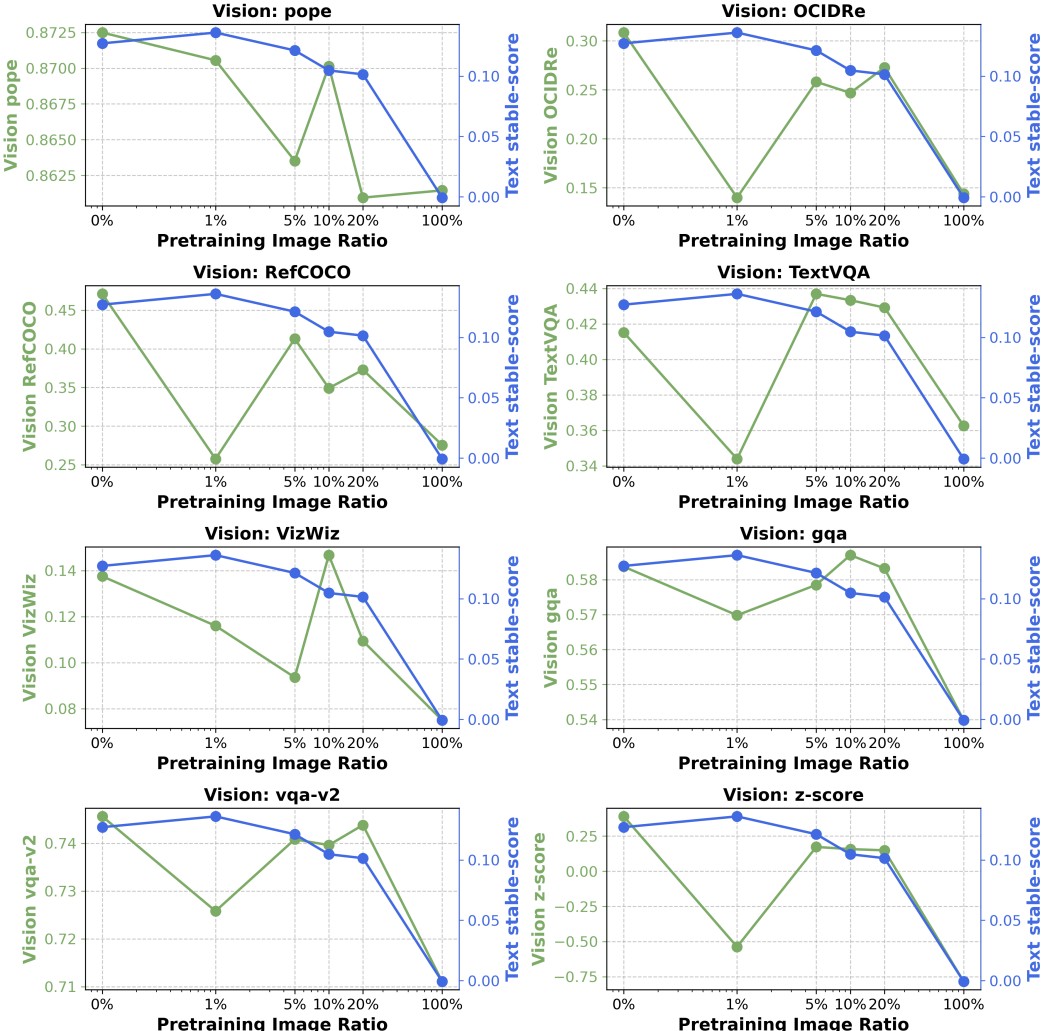

Figure 12: Evolution of the performance of the 1b model trained from scratch on vision benchmarks and text benchmarks as functions of the ratio of image caption data in the image pre-training phase.

## F.5 THE IMPACT OF INSTRUCTION TUNING DATA DURING PRE-TRAINING

Including instruction tuning data degrades VQA performance of the downstream model for all tested benchmarks. However, surprisingly, it improves the overall performance on downstream text tasks. We believe this is due to the instruction tuning data that help the model understanding question answering tasks.

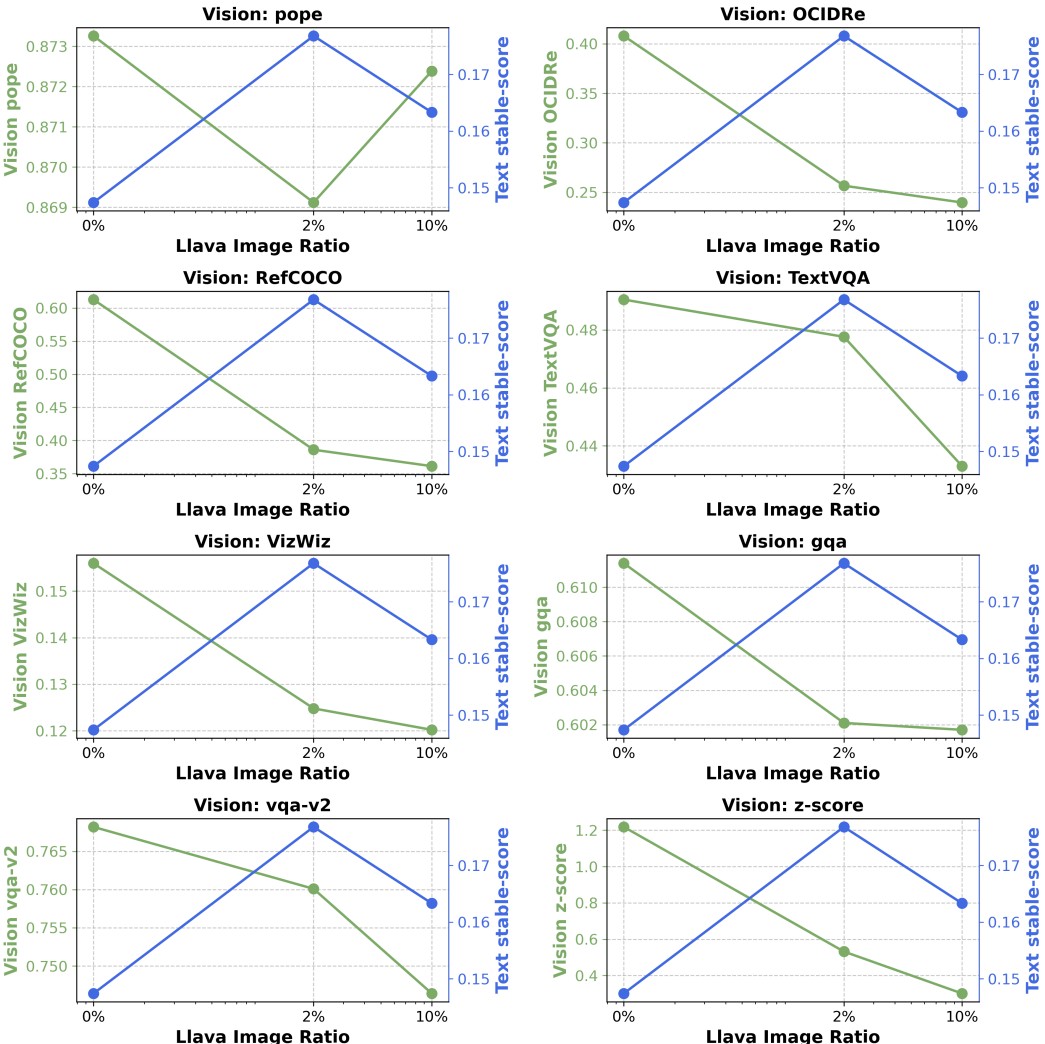

Figure 13: Evolution of the performance of the 1b model on vision benchmarks and text benchmarks as functions of the ratio of image instruction tuning in the image-text pre-training phase.

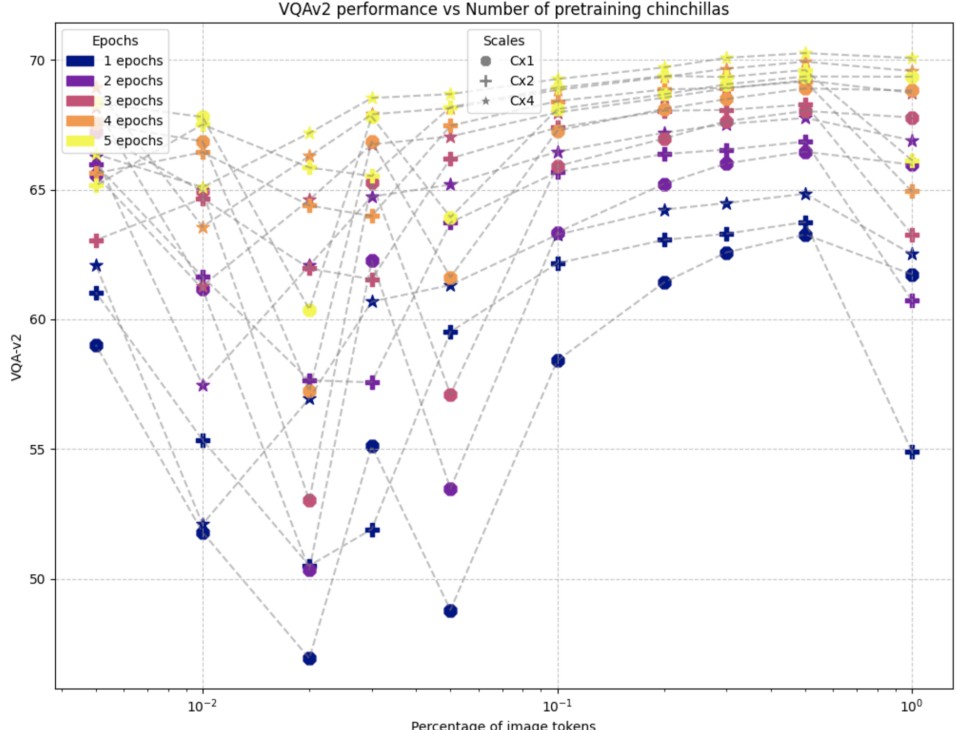

Figure 14: We train for {1,2,4} times the Chinchilla optimal number of tokens for 79M models. The above plots are taken for a 60% checkpoint, with VQA-v2 as the y-axis. Plots from other checkpoints look similar.

## G  TRAINING FOR HIGHER CHINCHILLA SCALES

In Figure 14, we train a 79M model for up to $4\times$ the Chinchilla scale, which is approximately an $80\times$ token multiplier. Overall, the trends for Cx1, Cx2, and Cx4 look quite similar. Consider, for example, the blue (epochs=1) trend lines for Cx1 (circle), Cx2 (cross), and Cx4 (star). We see that all three trend lines generally follow the same shape. The same can be said for the trend lines of the other epoch counts. We believe that this will be very interesting (albeit costly) to test at the 1B level to observe what role model size plays in these scale calculations.

## H  LARGER MODEL SCALES (7B)

In this paper, we mostly conducted experiments on 1B scales and below. One natural question to ask is whether the results will continue to hold at larger scales.

Experiments at the 7B scale can be costly, especially for pre-training. For text-only pre-training, we were able to leverage the pre-trained 7B checkpoints from DataComp-LM (Li et al., 2024b). However, DCLM-7B[4] was set to a schedule of 4.3T tokens but trained for only 2.3T tokens then cooled down. As such, the checkpoints for the 60%, 80%, and 100% runs do not exist. To test if our main claim still holds true in this modified setting, we train and compare the following models:

1. Introduce images (at a 90:10 text:image ratio) around 50% of the way through pre-training for a DCLM-7B checkpoint (2.3T tokens at a 4.3T token schedule). This is done at 1x Chinchilla tokens (140B tokens). Then fine-tune with LLaVA data for 4 epochs.

2. Take the final cooled-down DCLM-7B checkpoint (2.3T tokens cooled-down for 270B more tokens). For a fair token-matched comparison, we continue training with just text at the

---
[4] https://huggingface.co/apple/DCLM-7B

same token-matched scale as the model above (i.e. 1x Chinchilla tokens). Then fine-tune with LLaVA data for 4 epochs.

Here, model 1 is basically the method that we are proposing (i.e., adding images to the pre-trained checkpoint before it's fully cooled down), and model 2 is very similar to how many VLMs are conventionally trained.

Our results are shown in Table 3 below:

| Dataset | Model 1 (with image data in pre-training) | Model 2 (no image data in pre-training) |
|---|---|---|
| VQAv2 | 76.39 | 74.69 |
| GQA | 61.08 | 60.6 |
| VizWiz | 10.88 | 11.69 |
| TextVQA | 0.46828 | 0.41624 |
| RefCOCO | 0.64168 | 0.58889 |
| OCID | 0.46107 | 0.39696 |
| POPE | 0.87478 | 0.87113 |

Table 3: Performance of 7B models on various vision-language tasks. We see that model 1 (with images in pre-training) generally outperforms model 2 (no images in pre-training). This supports the findings that we observed in the smaller scale experiments.

## I    WHAT DO THE RESULTS LOOK LIKE IF WE UNFREEZE ALL THE WEIGHTS?

There has been a debate about whether or not to freeze the image encoders when training VLMs. Certain papers such as Prismatic (Karamcheti et al., 2024) claim that the performance is better if the encoders are frozen, whereas other papers such as Cambrian (Tong et al., 2024) and PaliGemma (Beyer et al., 2024) claim the opposite.

We conduct a few experiments to test this ourselves. Our results on the frozen vs unfrozen were conducted on 79M models. We report our results in Table 4. As the results for the unfrozen model were significantly worse across the board, we decided to stick with frozen image encoders for the rest of our experiments.

## J    ABLATIONS

### J.1    EFFECT OF IMAGE DATASET

In Table 5 we ablate over the caption dataset we use. We find that DataComp-DR is the best for both vision and text, though the gap is larger in text than in vision. This was one key reason we decided to use DataComp-DR for the majority of our experiments.

Table 4: All results are taken with the mix of DataComp-DR caption data and DCLM text data and fine-tuned on 3 epochs. Across multiple settings, the unfrozen models perform worse in general.

| Checkpoint | Frozen? | Text-Image Ratio | VQA-v2 score |
|---|---|---|---|
| 80% | yes | 5% | 58.71 |
| 80% | yes | 30% | 60.22 |
| 80% | yes | 50% | 60.85 |
| 80% | no | 5% | **68.55** |
| 80% | no | 30% | **69.42** |
| 80% | no | 50% | **71.0** |

Table 5: Effect of image dataset

| Dataset Mix | Vision Stable Score | Text Stable Score |
|---|---|---|
| **DataComp** | 0.4603 | 0.1168 |
| **DataComp-DR** | **0.4607** | **0.1503** |
| **CC12M** | 0.4556 | 0.1298 |
| **ShutterStock** | 0.4518 | 0.1310 |

Table 6: Effect of image encoder

| Image Encoder | Vision Stable Score | Text Stable Score |
|---|---|---|
| **SigLIP** | 0.4607 | **0.1503** |
| **SigLIP + DINO** | **0.4696** | 0.1347 |
| **Patch Projection** | 0.1564 | 0.1503 |

Table 7: All models above were trained from a 20% checkpoint on a mix of DataComp-DR caption data and DCLM text data.

| Random Seed | Image Ratio | FT epochs | VQA-v2 score |
|---|---|---|---|
| 7 | 1% | 1 | 73.83 |
| 7 | 1% | 2 | 75.39 |
| 7 | 5% | 1 | 75.36 |
| 7 | 5% | 2 | 76.39 |
| 7 | 10% | 1 | 75.39 |
| 7 | 10% | 2 | 76.5 |
| 365 | 1% | 1 | 74.46 |
| 365 | 1% | 2 | 75.9 |
| 365 | 5% | 1 | 75.39 |
| 365 | 5% | 2 | 76.47 |
| 365 | 10% | 1 | 75.55 |
| 365 | 10% | 2 | 76.58 |

## J.2 EFFECT OF IMAGE ENCODER

In Table 6 we ablate over image encoders. Here, we observe that SigLIP and DINO-SigLIP are quite close to each other for vision, with DINO-SigLIP narrowly edging out SigLIP. However, SigLIP is better by a large margin in text. We hence select SigLIP for the majority of our experiments.

## K RANDOM SEEDS

One question that arises when comparing our different experiments is to what extent a small variation of the result is significant. To control this factor, we experiment with different random seeds. The resulting variation of the result, shown in Table 7 below, is representative of the variability of our pipeline. Overall, we observe from Table 7 that the variation across seeds is quite small, although the variation may get slightly larger as we go to smaller image ratios such as 1%.

## L LIMITATIONS AND FUTURE WORK

While we strive to be thorough in our experiments, the space of exploration of multi-modal transformers is large and costly to cover. Many promising directions remain to be explored in this area.

- **Interleaved images** – This is something several recent papers have done (Laurençon et al., 2024; Awadalla et al., 2024; McKinzie et al., 2024). There are currently also several datasets available to use, such as Awadalla et al. (2024) and Laurençon et al. (2023). It is possible that using interleaved images will lead to new interesting results. To better handle this usage, we would need to re-think our image encoding scheme to reduce the sequence length of a sample counting several images.

- **Higher token multipliers for 1B models** – Right now, we have experiments with higher token multipliers (up to 4 times the Chinchilla (Hoffmann et al., 2022) optimal) for 79M models (Section G). However, we do not have these for 1B, as they can be costly to run. It

would also be interesting to go beyond 4x Chinchilla, as this would bring our training a lot closer to full pre-training as opposed to the cooldown that we are currently doing.

- **Scaling laws** – Through our experimentation at 79M parameter scale and 1B parameter scale, we discovered large differences in the model behavior and results through similar training pipelines. It might be a sign that scaling laws of VLMs are more complex than those of LLMs. More experimentation across model sizes would be needed to confirm if these behavioral shifts persist across scales and data, uncovering the unique scaling laws of Vision-Language Models (VLMs) compared to Large Language Models (LLMs).

- **Image resolution training** – This as well as other tricks from PaliGemma (Beyer et al., 2024) contribute to improve performance. However, they add complexity to the model and training pipeline, departing from the mainstream VLM training recipe. While we do not study the compounding effect these choices could have on the factors that we study, we believe these to be orthogonal improvement directions.

- **PrefixLM** – Using PrefixLM allows the model to attend bidirectionally over image patches and the task definition. This can possibly interplay with other factors such as freezing vs unfreezing the model, ultimately affecting the downstream performance after pre-training. By bringing part of the model closer to the usual ViT architecture, this could unlock some ability of the model to better capture the image input information, and potentially remove the need for a pre-trained image encoder.

## M  BENCHMARK RESULTS

To provide further analysis, we detail below the result per benchmark at different stages of our training process. Most text benchmarks see a decreasing score as the model is trained on images and then fine-tuned for visual question answering. Interestingly, AGIEval score actually increases with image training, while PubMedQA sees a sharp decrease in the score.

| Metric | Ours | Ours (No FT) | Base 80% | Base 100% | LLaMA 3.2 1B | Qwen 2.5 1.5B |
|---|---|---|---|---|---|---|
| AGIEval | 32.5 | 27.7 | 19.4 | 28.2 | 21.4 | 63.6 |
| ARC Easy | 59.3 | 72.3 | 74.7 | 77.1 | 69.5 | 80.6 |
| BigBench CC | 27.2 | 30.1 | 40.8 | 47.6 | 27.2 | 57.3 |
| BigBench CS | 35.8 | 46.2 | 44.6 | 46.7 | 46.5 | 56.5 |
| COPA | 71.0 | 83.0 | 86.0 | 92.0 | 83.0 | 85.0 |
| HellaSwag | 58.2 | 66.5 | 69.2 | 72.8 | 65.1 | 67.78 |
| MathQA | 22.9 | 25.9 | 26.9 | 27.3 | 30.52 | 40.84 |
| PIQA | 70.5 | 74.4 | 76.6 | 79.1 | 76.0 | 76.22 |
| PubMedQA | 35.0 | 38.6 | 66.2 | 69.6 | 65.8 | 66.6 |
| Stable-Score | 15.44 | 21.26 | 25.73 | 29.67 | 23.52 | 35.68 |

Table 8: Benchmark results at different stages of the training process. "Ours" is our model trained with 10% image captioning and 90% text before 4 epochs of LLaVA fine-tuning.

To help provide a qualitative assessment of our model performance, we also produce some data samples from VQAv2 with different question types. These samples are available at Zenodo.

## N  COMPARISON WITH PRIOR SOTA MODELS

While our primary focus in this paper is on studying training recipes, grounding our findings with a comparison to prior state-of-the-art (SOTA) visual language models (VLMs) provides valuable context and strengthens the validity of our results. Below, we present a comparison of our results with those reported by Prismatic VLM and PaliGemma. Although these models are larger, we were unable to find similarly sized models reporting the same set of evaluations as ours. We also plan to evaluate the Qwen-VL-2B-Instruct model and include it in this comparison.

| Metric | Ours | Prismatic 7B | PaliGemma 3B |
|---|---|---|---|
| TextVQA | 49.05 | 51.78 | $73.2 \pm 0.2$ |
| RefCOCO | 61.29 | 73.62 | $77.9 \pm 0.1$ |
| POPE | 87.33 | 88.28 | 87.0 |
| GQA | 61.11 | 64.16 | $67.0 \pm 0.3$ |
| VQAv2 | 76.82 | 79.05 | $85.6 \pm 0.2$ |
| OCID | 40.80 | 50.56 | N/A |
| Stable Score (No OCID) | 47.13 | 51.38 | 58.14 |
| Stable Score (With OCID) | 46.08 | 51.25 | N/A |

Table 9: Comparison of evaluation metrics between our model, Prismatic 7B, and PaliGemma 3B. "Ours" is our model trained with 10% image captioning and 90% text before 4 epochs of LLaVA fine-tuning.

