# OpenReview forum: "Should VLMs be Pre-trained with Image Data?"
_ICLR.cc/2025/Conference — ICLR 2025 Poster_

### Official Review · Reviewer_sMqo · 2024-10-21

**Soundness:** 3
**Presentation:** 4
**Contribution:** 4
**Rating:** 6
**Confidence:** 4

**Summary:**

This paper investigates the effect of image-text pretraining in large vision-language models using a one-stage approach, in contrast to the typical two-stage method where a text-only language model is adapted into a vision-language model. The study examines various factors during pretraining, such as the amount of text-only pretraining data, and the quantity, type, and ratio of image pretraining data. Specifically, after text-only training, the authors investigate the impact of when to start image-text training, the ratio of image-text pairs when performing image-text training, the impact of instruction-tuning data, and the finetuning epochs on image instruction tuning. After each factor is implemented, the model is followed by image instruction-based tuning, and each factor is then evaluated on VQA and text-only QA tasks. Experiments are conducted on a 1B model, with 28B tokens following the scaling law of (parameters x 20). Interesting findings in multimodal pretraining are deduced.

**Strengths:**

Generally, many strong insights that can advance the research of vision-language pretraining and Multimodal LLMs are deduced.

- I find that the insight of continued training at 80% preferable to training from the 100% pretrained text model, with a 90-10 image-text ratio, very interesting and will likely have a good impact on the community.

- The training from scratch insight (3.3) is also very interesting. This will highlight the importance of having a strong text-only model before interfering images. Interestingly, many old (2020-2022 century) vision-language models such as OSCAR, UNITER...etc, were following this "from-scratch" paradigm.

- I also find the insight on image instruction finetuning to also be degrading text-only task performance (3.5), interesting. This allows the community to deduce that when multimodal LLMs are instructed with text-only, they dont perform at their best. This could be helpful for systems to select the non-finetuned model when the prompt does not include an image.

**Weaknesses:**

- [W1] My main problem is that these insights are only demonstrated on QA and VQA tasks, which I dont find enough for deducing generalizable insights. Good VQA performance does not necessarily reflect general performance. There are a suit of benchmarks (e.g., those used in LLava) used to evaluate the vision-language models. At least include other vision-language tasks such as image captioning and its variations? Visual entailment? VQA with Textual Explanations (VQA-X) and visual entailment with explanations (e-SNLI-VE) can also be good dataset for testing.

- [W2] Something in Figure 5 caught my attention. When the image ratio is 0% (as if no image data is added, and we continue training on text-only data, as in a normal text-only LLM), the VQA performance is already around 75%, even though the model has seen no images. This makes me wonder that the model is answering questions without really looking at the image (this is a well-known bias problem in VQA models, presented in [R1]). Therefore, i am afraid these results may reflect biased models. I wonder what the results are without using any images? What about other VQA reasoning datasets such as A-OK-VQA?

- [W3] The authors mention in L739 in the appendix: “This time, we also make sure to mask out any system prompt and human question, only keeping the model responses unmasked.” If no question is there in the image instruction tuning stage, then this is not *instruction* finetuning (this is just a different dataset of image-text pretraining, similar form to the DataCompDR-1B). What is the point of conducting *instruction* tuning then? Im not sure exactly what is the rationale behind masking out the question from the instruction?

Other minor points (do not affect my decision):

- How is the image-text training on DataCompDR-1B dataset done? Is it to generate the caption autoregressively? Is it via Masked VL modelling? The authors dont mention this. I understand from section C that the authors dont use prefixLM (no image-image attention), and only casual modeling and text-image attention. Therefore, I assume that they predict the caption autoregressively? If so, please mention this, as this is not a trivial point, since many works in the literature use different VL pretraining objectives.

- In 3.4, when x% of image instruction tuning data is added during the image-caption training, is the same data repeated during the image instruction finetuning stage?

- L182-187, I feel this should go to the related work. The authors describe other works here.


[R1] RUBi: Reducing Unimodal Biases for Visual Question Answering

**Questions:**

My biggest problem is W1, and im afraid no general conclusions can be drawn. If time permits, I would like that the authors verify their insights on at least another task (e.g., captioning, VQA-X, e-SNLI-VE). I will increase my score if the authors address this concern. W2 is also concerning.

---

> ### Author Response · Authors · 2024-11-23
>
> Thank you for your comprehensive review and for highlighting the broader implications of our findings. We're particularly glad you recognized how our results on the 80% continued training approach and the 90-10 image-text ratio could impact future VLM development practices.
> Your historical perspective connecting our "from-scratch" findings to earlier vision-language models (2020-2022) adds valuable context.
>
> # Main Response:
>
> ## Additional Evaluations [W1]
> > My main problem is that these insights are only demonstrated on QA and VQA tasks, which I dont find enough for deducing generalizable insights. Good VQA performance does not necessarily reflect general performance. There are a suit of benchmarks (e.g., those used in LLava) used to evaluate the vision-language models. At least include other vision-language tasks such as image captioning and its variations? Visual entailment? VQA with Textual Explanations (VQA-X) and visual entailment with explanations (e-SNLI-VE) can also be good dataset for testing.
>
> This is a good point – we agree that using only VQA is a bit restrictive.
>
> 1. Our evaluations also include other metrics aside from general VQA in our aggregate metric (which we called “stable_score”). Specifically, we considered a variety of tasks such as TextVQA (document understanding and OCR), RefCOCO (object localization) and OCID-Ref (cluttered object localization). Our current evaluation contains at least one benchmark from each of the categories listed in the Qwen2-VL paper (except for video/multilingual). See Appendix E for a detailed list of our tasks.
> 2. Nonetheless, we agree that using a larger set of benchmarks would be better. To fix this, we add our scores on MMMU below. We are also evaluating on other datasets and will add them over the next few days as soon as the results come in.
> | Checkpoint | MMMU Accuracy |
> |------------|-----|
> | 90-10 text-image ratio, 100% checkpoint, 4 epochs | 0.3066 |
> | 90-10 text-image ratio, 80% checkpoint, 4 epochs | 0.32 |
> | 90-10 text-image ratio, 60% checkpoint, 4 epochs | 0.327 |
> | 90-10 text-image ratio, 40% checkpoint, 4 epochs | 0.3533 |
> | 90-10 text-image ratio, 20% checkpoint, 4 epochs | 0.24667 |
> | 90-10 text-image ratio, 0% checkpoint, 4 epochs | 0.29333 |
> |PaliGemma-3B-mix-448			          | 0.336 |
> |Qwen2-VL-2B					          | 0.423 |
> |LLaVA-Next-Mistral-7B			          | 0.37 |
>
> The results shown are for our model initialized from the different text pretraining checkpoints, then trained on image text with a mix of 10% image captioning, 90% text, then finetuned for 4 epochs. The compared models, PaliGemma, Qwen-VL have public results for MMMU (https://huggingface.co/spaces/opencompass/open_vlm_leaderboard) we recomputed them to make sure our evaluation pipeline does not make a difference and computed the results in the same way for LLaVA-Next-Mistral. Surprisingly, and as opposed to our previous results, they do not show the 80% checkpoint to be best but it is still stronger than the 100% checkpoint.
>
> ## Performance of 0\% image data model [W2]
> >  Something in Figure 5 caught my attention. When the image ratio is 0% (as if no image data is added, and we continue training on text-only data, as in a normal text-only LLM), the VQA performance is already around 75%, even though the model has seen no images. This makes me wonder that the model is answering questions without really looking at the image (this is a well-known bias problem in VQA models, presented in [R1]). Therefore, i am afraid these results may reflect biased models. I wonder what the results are without using any images? What about other VQA reasoning datasets such as A-OK-VQA?
>
> To clarify, the 0\% here refers to the 0\% images used for pre-training. There is always a fine-tuning stage for all models, so the 75\% that you’re seeing in Figure 5 is after fine-tuning. For more details, see the green “Setup” boxes – we finetune for 4 epochs. In other words, we never use a model that is trained only on text.
>
> ## Masking [W3]
> > The authors mention in L739 in the appendix: “This time, we also make sure to mask out any system prompt and human question, only keeping the model responses unmasked.” If no question is there in the image instruction tuning stage, then this is not instruction finetuning (this is just a different dataset of image-text pretraining, similar form to the DataCompDR-1B). What is the point of conducting instruction tuning then? Im not sure exactly what is the rationale behind masking out the question from the instruction?
>
> Thank you for noticing this imprecision. We will clarify this point in the main paper. Here by “masked”, we meant to say masked in the loss computation. It is given without masking as a model input.

---

> > ### Author Response · Authors · 2024-11-23
> >
> > # Minor Points
> >
> > > How is the image-text training on DataCompDR-1B dataset done? Is it to generate the caption autoregressively? Is it via Masked VL modelling? The authors dont mention this. I understand from section C that the authors dont use prefixLM (no image-image attention), and only casual modeling and text-image attention. Therefore, I assume that they predict the caption autoregressively? If so, please mention this, as this is not a trivial point, since many works in the literature use different VL pretraining objectives.
> >
> > The model is trained to predict the caption auto-regressively. We do not use Masked VL. Thank you for noticing this, we will include this important information in section 2.1.2.
> >
> > > In 3.4, when x% of image instruction tuning data is added during the image-caption training, is the same data repeated during the image instruction finetuning stage?
> >
> > Yes this is the same data but it represents a fraction of an epoch at the end of training.

---

> ### Comment · Reviewer_sMqo · 2024-11-23
> **Response by reviwer sMqo**
>
> I thank the authors for their response and additional experiments. However, can the authors comment on the MMMU results? As from what I see, they do not agree with the hypothesis mention in the paper, as it seems that the 40% checkpoint works best (in the main paper, it was 80% that the authors claimed that worked best). Additionally, the results shown are for what model? or are they for the method in Figure 2? or PaliGemma? Or Qwen2-VL-2B? What ratio and what checkpoints are these other models (PaliGemma, Qwen2-VL-2B, LLaVA-Next-Mistral-7B) using? I am not sure why the authors paste this table with no discussion.
>
> In general, i see that my doubt is correct, the hypothesis is only valid for VQA tasks, and it is not generizable to other tasks. I am not sure about the impact this paper will have in that case.

---

> ### Author Response · Authors · 2024-11-23
>
> Thank you for your interest and this follow-up question. We should have explained and discussed these results further. We are also surprised that the MMMU results show better performance from the 40% checkpoint and did not have the time to explore the reason why further at this time.
>
> A few points we want to raise regarding this:
> 1. While these new results do not support our claim that the 80% checkpoint is the best, it still shows better results than the 100% checkpoint (supporting our claim). Moreover, considering all benchmarks, the 80% checkpoint is still the best choice for a generalist model.
>
> 2. In Appendix F, there are also other tasks like VizWiz where the 80\% checkpoint doesn't perform the best. This is why we report an aggregate metric such as stable score, individual benchmark results might depart from the average performance. There are some benchmarks (like MMMU as we're seeing) where it doesn't perform the best (and that's totally expected). The main advantage of using some aggregate metric like stable score is so that we don't overindex on a single task like this.
> - Now this brings the question of how well-designed our aggregate metric is, and whether the current set of tasks we're considering is too narrow. We tried our best to consider a variety of tasks, but we agree that it would be a more representative metric if we added a more diverse set of tasks. We are currently working on adding more tasks such as captioning and entailment and will let you know once we get more results.
>
> ---
>
>
>
> The results shown are for our model initialized from the different text pretraining checkpoints trained on image text with a mix of 10% image captioning, 90% text, then finetuned for 4 epochs.
> The compared models, PaliGemma, Qwen-VL have public results for MMMU (https://huggingface.co/spaces/opencompass/open_vlm_leaderboard) we recomputed them to make sure our evaluation pipeline does not make a difference and computed the results in the same way for LLaVA-Next-Mistral. These models use the fully trained initial LLM checkpoint.
>
> We would also like to point out that we did include non-VQA benchmarks in our published evaluations such as OCID-ref and RefCOCO that are localization benchmarks evaluated on intersection over union of bounding boxes. These results also show better performance from the 80% checkpoint.

---

> ### Comment · Reviewer_sMqo · 2024-11-25
> **response by Reviewer sMqo**
>
> I thank the authors for their response. I think the authors present an interesting finding that will likely have an impact on VLMs. This finding basically says to introduce image pretraining for 10% of the time, earlier than expected as per the scaling law (at 11.2B and 22.4B tokens for 40% and 80%, respectively). The only reason I am not raising my score and giving this paper an accept (score of 8) is because 1) this finding may depend on the VLM and architecture (as the authors showed results on their own architecture), 2) (my initial concern) there is no clear rule of when to actually introduce the image tokens. Sometimes 40% is better for a set of tasks, sometimes 80% is better for another set of tasks. Although both lead to increased performance, i still find it necessary to have some guidelines for researchers to follow (although i understand that its hard to have one rule that works on all). Therefore, I wish to stay with my score as a borderline acceptance.

---

### Official Review · Reviewer_ans3 · 2024-10-31

**Soundness:** 3
**Presentation:** 3
**Contribution:** 2
**Rating:** 5
**Confidence:** 4

**Summary:**

This paper studied when to integrate image tokens to the training process of LVLM. The authors trained models spanning various datasets, scales, image-text ratios, and amount of pre-training done before introducing vision tokens. It experimentally found that incorporating image data during pre-training generally helps and the fraction of visual data and time of introducing visual instruction tuning are also important.

**Strengths:**

1. This paper systematically studies the time of introducing visual tokens during language pre-training, image-text pre-training and instruction finetuning.
2. It demonstrates several key findings such as incorporating image data during pre-training, and the time of introducing instruction fine-tuning image tokens.
3. Compared with fully pre-trained model, introducing visual tokens 80% of the way through pre-training results in a 2% average improvement in VQA tasks.

**Weaknesses:**

1. This study is very limited to the parameter size of the LLM. It seems hard to generalize to different size of LLMs. It seems this paper adopted 1B LLM, commonly used 7B and 13B LLM should also be studied. How do authors expect the findings might change at different model scales?
2. Some findings are already well-studied, such as mixing together instruction fine-tuning during the image-text pre-training can hurt the model.
3. Current LVLM starts with a fully pre-trained LLM. But it seems not many strong LLMs can release their weight at 80%. What are the practical implications of this paper's' findings given the limited availability of partially trained LLM checkpoints? Could the authors explore ways to approximate the approach using fully trained models?
4. Would like to see the final model trained by this paper compared with existing LVLMs such as mentioned in paper's Table 1 or well-recognized models such as LLaVA-Next, LLaVA-One-Vision etc.

**Questions:**

1. This is a clarification question: When incorporate image at a early time, does that mean more visual data is trained or seen by model?
2. I think the name of "LLaVa" should be "LLaVA".

---

> ### Author Response · Authors · 2024-11-23
>
> Thank you for your detailed review of our experimental work on VLM training optimization. We're pleased that you found our extensive ablation studies and empirical findings to be well-supported and meaningful. Your recognition of the practical significance of our results is greatly appreciated.
>
> # Main Response:
>
> ## Scaling (Weakness #1):
> > This study is very limited to the parameter size of the LLM. It seems hard to generalize to different size of LLMs. It seems this paper adopted 1B LLM, commonly used 7B and 13B LLM should also be studied. How do authors expect the findings might change at different model scales?
>
>
> We performed a similar set of experiments with an LLM of 154 million parameters and another with 411 million parameters and found consistent patterns, with the exception that smaller models’ downstream performance require more than 4 epochs for the performance to degrade. The trend also shows that the difference in downstream performance between the models initialized from the 80% text training and 100% is decreasing as the model size increases.
>
> Because our findings mostly hold from 154M to 411M to 1B, we believe that it will continue in larger scales. We attach our scaling plots in this anonymous link: https://zenodo.org/records/14207627
>
> We are currently running a set of experiments at the 7B scale. However, there’s a chance these experiments may not finish in time for this discussion period. We wanted to get this response out first so that we could begin author-reviewer discussions, but we will share the 7B results as soon as we get them.
>
>
> ## Availability of checkpoints (Weakness #3):
> > Current LVLM starts with a fully pre-trained LLM. But it seems not many strong LLMs can release their weight at 80%. What are the practical implications of this paper's' findings given the limited availability of partially trained LLM checkpoints? Could the authors explore ways to approximate the approach using fully trained models?
>
>
> We agree that requiring partially trained checkpoints may limit immediate applicability of this paper’s findings. However, we hope this work will encourage open-model contributors to release more than just their final checkpoints in the future. Intermediate checkpoints are likely stored by model developers and could be released if there is sufficient demand, furthering open research. Publishing papers such as ours which shows the value of intermediate checkpoints can help push the field towards this direction.
>
> Moreover, while our findings are directly relevant to labs with access to internal checkpoints, they also provide broader insights into the training dynamics of language models, contributing to the field’s understanding. Importantly, our experiments at smaller scales demonstrate that these insights are applicable even to those working on smaller models, ensuring accessibility beyond frontier labs.
>
> We also appreciate the suggestion to explore approximations using fully trained models, which is an interesting direction for future research.
>
>
> ## Comparisons to existing models (Weakness #4):
> > Would like to see the final model trained by this paper compared with existing LVLMs such as mentioned in paper's Table 1 or well-recognized models such as LLaVA-Next, LLaVA-One-Vision etc.
>
> While our aim is to study training recipes in this paper, we agree that grounding the results with prior SOTA VLM results comparison will strengthen our results and be more convincing about the validity of our results. Find below a comparison of the results reported by Prismatic VLM and PaliGemma. Note that these are larger models but we could not find similar sized models that report the same set of evaluations as ours. We will run evaluation on the Qwen-VL-2B-Instruct model and add it to this table.
>
> | Metric                      | Ours (80\% checkpoint) | Prismatic 7b      | PaliGemma 3b           |
> |-----------------------------|---------------------|-------------------|------------------------|
> | **TextVQA**                | 49.05              | 51.78            | 73.2 ±0.2             |
> | **RefCOCO**                | 61.29              | 73.62            | 77.9 ±0.1             |
> | **POPE**                   | 87.33              | 88.28            | 87.0                  |
> | **GQA**                    | 61.11              | 64.16            | 67.0 ±0.3             |
> | **VQAv2**                  | 76.82              | 79.05            | 85.6 ±0.2             |
> | **OCID**                   | 40.80              | 50.56            |  N/A              |
> | **Stable Score (No OCID)** | 47.13              | 51.38            | 58.14                 |
> | **Stable Score (With OCID)**| 46.08             | 51.25            | N/A                   |
> ---

---

> > ### Author Response · Authors · 2024-11-23
> >
> > # Minor Points:
> > > Some findings are already well-studied, such as mixing together instruction fine-tuning during the image-text pre-training can hurt the model.
> >
> > Yes, and we still think these findings are worth reporting:
> > - We confirm empirical findings, which sometimes there are some contradictions in, so it is worth reporting that we are able to replicate known findings.
> > - The main focus of our paper is on including images in pre-training and exploring VLM pre-training dynamics, which is something new.
> >
> > > When incorporate image at a early time, does that mean more visual data is trained or seen by model?
> >
> > We make sure to keep the amount of training consistent (28B tokens or approximately 1x Chinchilla tokens), so whether we incorporate image at the 20% checkpoint or at the 80% checkpoint, it is always training on the same number of visual tokens.
> >
> > Note, however, that if we compare with the standard LLaVA pipeline which is initialized from a text-only model, then yes, our model will see more visual data because we incorporated images at an earlier time. We evaluated the models after different number of fine-tuning data and noticed a saturation point at 4 epochs. We did not notice that pre-training with images would lower the finetuning saturation point.
> >
> > For reference, we also have a plot showing {1,2,4}x Chinchilla in Appendix G.
> >
> >
> > > I think the name of "LLaVa" should be "LLaVA".
> >
> > Thank you for pointing this out! We agree it's good to be consistent here. We will fix it in our updated PDF.

---

> ### Author Response · Authors · 2024-12-02
>
> Hi reviewer ans3,
>
> # 7B experiments
> We have trained a 7B model, and we are happy to share that our method indeed scales well to the 7B scale. This confirms the main finding of our paper, which is that pre-training with a mixture of image and text data allows models to perform better on downstream tasks.
>
> More specifically, we compared two 7B models as follows:
> 1. Introduce images (at a 90:10 text:image ratio) \~50\% of the way through pre-training for a DCLM-7B checkpoint (2.3T tokens at a 4.3T token schedule). This is done at 1x Chinchilla tokens (~140B tokens). Then fine-tune with Llava for 4 epochs.
> 2. Take the final cooled-down DCLM-7B checkpoint (2.3T tokens cooled-down for 270B more tokens). For a fair token-matched comparison, we continue training with just text at the same token-matched scale as the model above (i.e. 1x Chinchilla tokens). Then fine-tune with Llava for 4 epochs.
>
> Here, model 1 is basically what we're proposing, and model 2 is very similar to how many VLMs are conventionally trained.
>
>
> Our results are as follows:
>
> | Dataset  | Model 1 (with image data in pre-training) | Model 2 (no image data in pre-training) |
> |----------|----------|----------|
> | VQAv2    | **76.39** | 74.69    |
> | GQA      | **61.08** | 60.6     |
> | VizWiz   | 10.88    | **11.69** |
> | TextVQA  | **0.46828** | 0.41624  |
> | RefCOCO  | **0.64168** | 0.58889  |
> | OCID     | **0.46107** | 0.39696  |
> | POPE     | **0.87478** | 0.87113  |
>
> We hope that this gives you a bit more confidence in the validity and scalability of our results. If time permits, we will add even more 7B results to the final version.
>
>
>
> ---
> ..
> ---
>
> With this update, we hope we have addressed all of the issues that you have raised. We deeply appreciate your feedback and we’d like to explore whether there might be a chance to reevaluate the score in light of all the points we have addressed. If you have further questions or concerns that you feel is a major blocker for acceptance, please let us know.

---

### Official Review · Reviewer_7Ue5 · 2024-11-03

**Soundness:** 2
**Presentation:** 4
**Contribution:** 3
**Rating:** 5
**Confidence:** 4

**Summary:**

Authors investigate the common multi-step training procedure of VLMs, where first an LLM is trained, and then this LLM is further trained with image-text data to turn it into a VLM. They probe when and how to exactly add the image-text data into the training, and how to set the learning rate schedules. Also, they research the impact of setting different ratios of image/text tokens in the visual pretraining stage, and how to exactly use the image instruction data for the last last finetuning stage.

Their findings show that image-text data should be added during the cooldown phase of the LLM training, and not as a second step with a second warmup phase. They further find that adding instruction fine-tuning data during the pretraining stage hurts performance, and that 10-20% of tokens should be visual during the image-text pretraining stage.

**Strengths:**

The paper is very well presented visually and also very well written, as it is fluent to read and understandable. This combination of nice figures and well written text makes it well digestable. The authors highlight potential weaknesses or difficulties, e.g. in figure 4 highlighting the points that are trained with a different learning rate schedule.

The research question is interesting and valuable, since most VLM pretraining happens in (semi-)proprietary setups where details are hidden, or in setups were many training choices are not discussed since ablating them would be difficult.

The finding that image-text data should be injected earlier into the training process is relevant for many existing VLM training pipelines.

**Weaknesses:**

Note that these weaknesses are in descending order of importance.

1) The comparison to other VLMs is missing: The authors introduce a new metric “vision stable-score” as their main empirical evidence, but then evaluate only their own trained models on this metric. This makes comparing this work to other works difficult. Authors should show the benchmark numbers for some of the models in table 1. This work does not need to be top 1 in this table since the goal is not to train a SOTA model but to investigate training behaviour. However the comparison table is needed to evaluate how close this work is to the SOTA VLMs, because if the model is too far away, the findings might not relevant for SOTA models.

2) Similarly, the text stable score is very low, below 20% everywhere. This might be due to subtracting the random baseline. However there needs to be a comparison to the DCLM-1B text model before any image training, as well as maybe one more SOTA/popular LLM.

3) In 3.1 the peak learning rate is set to 3e-3 for the 100% checkpoint model. This choice is not argued for or ablated. The best peak learning rate in this chapter seems to be the green cosine 80% curve in figure 3 (around 1e-4 however it is hard to read the exact number from the graph), therefore authors should have at least tried this best peak learning rate also for their 100% checkpoint model. Otherwise, the risk remains that the main finding of the paper in chapter 3.1 can be just attributed to different hyperparameters.

4) Evaluating only on accuracy metric benchmarks has significant drawbacks, since only short and exactly matching answers can be correct. Considering that this model is trained on image-caption pairs, there should be at least a captioning benchmark added.

5) In the abstract: “a 1B model (retains) performance within 2% of our text-only baseline”. I could not easily find this in the paper. Assuming you compare the 80% model from figure 4 with ~15% text stable score and ~54% text arc easy. The text-only baseline should then be the DCLM-1B model without any image-text training and without image-text finetuning. However that model reports ~75% text arc easy results. So: Which models are you comparing to make this statement, on which metrics? If the text-only baseline is not the DCLM-1B model, why?

6) The decision to freeze the vision encoder was done on a 79M model. However there is no argument why this should hold for bigger models, it would be good to have at least one unfrozen run on the 1.4B model.

7) Training on interleaved image-text data is omitted in this work. This is a weakness acknowledged by the authors but it is reasonable to first evaluate the findings on image-text pairs and potentially in the future extend to interleaved data.

Summary: I believe that points 1 to 3 must be addressed before the paper can be published, because otherwise the scientific evidence for the findings is not strong enough.

**Questions:**

- The LLM is only trained for 28B tokens in this work, while it is originally trained for 4.3T tokens in the DCLM-1B work, a factor 150x more data. Authors argue with the scaling laws from the Chinchilla paper, which would suggest “20 times number of params” of tokens is enough. But then, why use a LLM that is trained on “3070 times number of params” tokens?

- How many (approximate) GPU hours were used for each experiment, or at least, how expensive is one run of the 1.4B model on 28B tokens? Appendix A shows GPU hours for the text-only training but not for the image-text training.

---

> ### Author Response · Authors · 2024-11-23
>
> Thank you for your thorough and thoughtful review of our work. We appreciate your positive feedback on the presentation and clarity of our findings. Your comments highlight the importance of our research in advancing the understanding of VLM training procedures, particularly in studying the optimal timing and approach for incorporating image-text data. We're especially glad you found value in our key finding about injecting image-text data during the LLM cooldown phase rather than as a separate training stage.
>
>
> # Main Responses
>
> 1. [W1] While our aim is to study training recipes in this paper, we agree that grounding the results with prior SOTA VLM results comparison will strengthen our results and be more convincing about the validity of our results. Find below a comparison of the results reported by Prismatic VLM and PaliGemma. Note that these are larger models but we could not find similar sized models that report the same set of evaluations as ours. We will run evaluation on the Qwen-VL-2B-Instruct model and add it to this table.
>
> | Metric                      | Ours (80% checkpoint, 4 FT epochs) | Prismatic 7b      | PaliGemma 3b           |
> |-----------------------------|---------------------|-------------------|------------------------|
> | **TextVQA**                | 49.05              | 51.78            | 73.2 ±0.2             |
> | **RefCOCO**                | 61.29              | 73.62            | 77.9 ±0.1             |
> | **POPE**                   | 87.33              | 88.28            | 87.0                  |
> | **GQA**                    | 61.11              | 64.16            | 67.0 ±0.3             |
> | **VQAv2**                  | 76.82              | 79.05            | 85.6 ±0.2             |
> | **OCID**                   | 40.80              | 50.56            |  N/A              |
> | **Stable Score (No OCID)** | 47.13              | 51.38            | 58.14                 |
> | **Stable Score (With OCID)**| 46.08             | 51.25            | N/A                   |
>
> 2. [W2] The text performance that we report is indeed fairly low and we only discussed the trends, neglecting their absolute values in the paper. We agree that absolute values and comparison with SOTA models is important. Thus, we report our results in the table below and compare them with the performance of SOTA baselines: LLama 3.2 which performs worse than the OpenLM model we used and Qwen 2.5 which performs significantly better but was released after the beginning of this work.
> The text scores are lowered by the image training process (Ours no FT is our model from the base 80% trained on image captioning and text). The performance is reduced by 4.5% after image training and an additional 5.5% after LLaVA fine-tuning, as shown in the table below. (Interestingly, AGIEval is the only text benchmark that sees an increase in score after image training and again after LLaVA finetuning.) To our knowledge other works do not report the text performance of the models after tuning them for image understanding.
>
> | Metric                                      | Ours (4 epochs LLaVA FT) | Ours (No FT) | Base 80% | Base 100% | Llama 3.2 1B | Qwen 2.5 1.5B |
> |--------------------------------------------|--------------------------|--------------|-----------|-----------|--------------|---------------|
> | **AGIEval**                         | 32.5                   | 27.7         | 19.4      | 28.2      | 21.4         | 63.6         |
> | **ARC Easy**                               | 59.3                   | 72.3        | 74.7     | 77.1     | 69.5        | 80.6         |
> | **BigBench Conceptual Combinations**       | 27.2                    | 30.1         | 40.8     | 47.6     | 27.2         | 57.3         |
> | **BigBench CS Algorithms**                 | 35.8                   | 46.2        | 44.6     | 46.7     | 46.5        | 56.5         |
> | **COPA**                                   | 71                   | 83        | 86     | 92     | 83        | 85         |
> | **HellaSwag**                              | 58.2                   | 66.5        | 69.2     | 72.8     | 65.1        | 67.78         |
> | **MathQA**                                 | 22.9                    | 25.9         | 26.9      | 27.3      | 30.52        | 40.84         |
> | **PIQA**                                   | 70.5                   | 74.4        | 76.6     | 79.1     | 76.0        | 76.22         |
> | **PubMedQA**                               | 35.0                    | 38.6         | 66.2     | 69.6     | 65.8        | 66.6         |
> | **Total Stable-Score**                     | 15.44                   | 21.26        | 25.73     | 29.67     | 23.52        | 35.68         |

---

> ### Author Response · Authors · 2024-11-23
>
> # Responses (cont.)
>
> 3. [W3] Thank you for raising this. We think it’s a very good point that will help strengthen our paper. We trained a new model using the maximum learning rate of the 80% checkpoint and the initial checkpoint at 100% of the text training and report the results below. It appeared that our choice of hyperparameters was not optimal since this one yields slightly better results. However, the results from the 80% checkpoints are still better, so these new results support our overall conclusion.
> | Model                          | Stable-Score Text | Stable-Score Vision |
> |--------------------------------|-------------------|---------------------|
> | 100% checkpoint                | 13.87            | 43.35              |
> | 80% checkpoint                 | 15.44            | 46.07              |
> | 100% checkpoint trained with 80% parameters | 14.29            | 45.61              |
>
>
> 4. [W4] We agree that evaluating only on accuracy has significant drawbacks. We would like to point out that we included tasks such as RefCOCO and OCID, which are bounding box prediction tasks that are evaluated as a percentage of results greater than a threshold which is not an exact match or multiple choice metric. It measures the percentage of produced bounding boxes with an intersection over union greater than a threshold.
>
>
> # Minor points
>
> 5. [W5] This statement from our paper is wrong, thank you for noticing this mistake, it will be removed. See our answer to your point W2 for more details.
>
> 6. [W6] From the PaliGemma paper, paragraph 3.2.1, it is reported that unfreezing the image encoder is better but it requires some adaptations of the learning rate schedule with two different schedules, one for the image encoder and one for the LLM. We leave this study that would involve tuning some additional hyperparameters to future work.
>
> ### Questions
>
> > The LLM is only trained for 28B tokens in this work, while it is originally trained for 4.3T tokens in the DCLM-1B work, a factor 150x more data. Authors argue with the scaling laws from the Chinchilla paper, which would suggest “20 times number of params” of tokens is enough. But then, why use a LLM that is trained on “3070 times number of params” tokens?
>
> It is true that better performance would be achieved with more training. When saying that  “20 times the number of parameters is enough”, we do not claim that it is enough to achieve the best performance, only that we believe it is enough to study trends in the different models performance. We cannot claim this number is optimal in any way in the context of image-text training, only that it is a reasonable training time to study model performance variations. We used the best LLM available that is widely overtrained as you noted because it does yield better performance and is closer to what practitioners would chose.
>
> > How many (approximate) GPU hours were used for each experiment, or at least, how expensive is one run of the 1.4B model on 28B tokens? Appendix A shows GPU hours for the text-only training but not for the image-text training.
>
> For a 28B tokens image-text training of a 1.4B parameter model, we used 2 nodes of 8 H100 for 4h15min.
> For a 4-epoch fine-tuning, we used either a node of 8 40GB A-100 for 10h40min or a node of 8 H100 for 2h20min.

---

> ### Comment · Reviewer_7Ue5 · 2024-11-25
>
> Thank you for your detailed responses. I have two follow-up questions:
>
> "[...] we included tasks such as RefCOCO and OCID, which are bounding box prediction tasks [...]" - how exactly are you running object detection on your VLM, are you finetuning an object detector head on top of your frozen features, or does your model include object detection capabilities? I could not find this in the paper or supplementary. So far as I understood it after finetuning on the Llava Instruction Data there is no further finetuning on the target sets.
>
> Statement "It does this while retaining performance within 2% of our text-only baseline on downstream text evaluations." "[...] will be removed". When are you planning to update the paper?

---

> ### Author Response · Authors · 2024-11-25
>
> Good questions!
>
> 1. Actually the `llava_v1_5_mix665k.json` subset of https://huggingface.co/datasets/liuhaotian/LLaVA-Instruct-150K already contains bounding box prediction as one of its tasks, so we do not need a separate finetuning step to do bounding box prediction.
>
> Specifically, Llava uses the prompt "Please provide the bounding box coordinate of the region this sentence describes:"
>
> For further exploration, please check out this short Jupyter notebook (anonymous): https://colab.research.google.com/drive/1ePSJsvGM9fOcPv9x4VWPcx3feN8jWZfB
>
> ---
> 2. Regarding updating the paper, we will probably do it after the review process. We understand that we could do it now, but we prefer to keep the paper in its original state so that reviewers and AC can see and compare when doing these discussions. We feel it might get a bit confusing if the reviewers are discussing something and then suddenly the point in discussion is no longer in the PDF anymore or it gets changed in the PDF suddenly.
>
> If you feel strongly towards us updating the paper now, please let us know, and we will gladly do so.

---

> > ### Comment · Reviewer_7Ue5 · 2024-11-26
> >
> > Thanks for the clarification about pretraining.
> >
> > The conference enables updating the paper. In my personal opinion it's better to update the paper now to reflect the feedback, especially if it leads to major changes like changes in the abstract.

---

> > > ### Author Response · Authors · 2024-11-27
> > >
> > > That makes sense. We've updated our PDF. Specifically, we edited the abstract to remove the incorrect sentence and edited the last sentence to highlight how we're evaluating on not just VQA tasks.

---

> ### Author Response · Authors · 2024-12-02
>
> Hi reviewer 7Ue5, we just wanted to let you know that we have posted a new overall response regarding some new updates we have during the rebuttal period.
>
> https://openreview.net/forum?id=Pj4Aid3XqL&noteId=5VHjy4YvN9
>
> This is mainly about the scalability (to 7B) and generalizability (to other benchmarks) of our work.
>
> ---
>
> We really appreciate the detailed feedback you gave to us. It is validating to hear you found our idea relevant, and we sincerely thank you for taking the time to share your knowledge and insights on our paper. We were wondering if there might be room to revisit the score, given that we feel our paper right now is much stronger than the paper we originally submitted. Ultimately we feel like we have shown enough results to scientifically prove our overall conclusion. If you have further questions or concerns that you feel is a major blocker for acceptance, please let us know.

---

### Official Review · Reviewer_Z7KL · 2024-11-04

**Soundness:** 3
**Presentation:** 3
**Contribution:** 3
**Rating:** 5
**Confidence:** 4

**Summary:**

Brief Summary: This is an experiment heavy paper tackling the question of how to best pre-train Vision-Language Models (VLMs). In particular, the authors investigate the best time to provide vision-text data to the existing LLM model, how much visual data should be provided to maintain scores on language-only tasks.

The experiments on multiple VQA datasets and text-benchmarks (noted in appendix E) suggest some key takeaways such as timing of introducing the vision data in the cooldown phase of the LLM training, the fraction of the vision data being limited compared to text data, and using instruction fine-tuning tokens for downstream tasks during fine-tuning is better.

**Strengths:**

Pros:

1. This is an experiment heavy paper aiming to empirically provide a guideline for training with image data for image-language downstream tasks. The authors suggest they have performed over 300 runs (some on 79M some on 1B model) which is generally sufficient.

2. The takeaways make sense. The improvements by following the guidelines are significant (~2% on vqa benchmarks).

3. The authors provide heavy ablations, the experiments are well thought out in general.

**Weaknesses:**

Cons:

1. The authors should provide some qualitative analysis, of where the model scoring lower misses compared to other models. Currently, it is not directly clear what kind of questions the 2% drop come from.

2. One concern is that the chosen VQA benchmark is a bit restrictive. The authors should chose a larger set of evaluation benchmark such as MMMU, OCR-Bench to name a few (Qwen2-VL has a good set of possible evaluation benchmarks [Ref1]).

3. Related to previous point, the authors should also include some caption/description as well as OCR heavy datasets as well.

4. (Minor) The provided methodology is shown on 1B models but not on larger models such as 7B and 70B. While in general it is expected the findings will carry forward, an explicit experiment would be better. However, it might be prohibitively expensive.

5. (Minor) One issue with the guidelines is that these can be only applied at training time by the authors of the original work. Thus, if someone else wants to improve an existing VLM, it requires the original authors to have open-sourced their checkpoints including the optimizers weights and schedule which may be difficult to obtain.

[Ref1]: Wang, Peng, Shuai Bai, Sinan Tan, Shijie Wang, Zhihao Fan, Jinze Bai, Keqin Chen et al. "Qwen2-vl: Enhancing vision-language model's perception of the world at any resolution." arXiv preprint arXiv:2409.12191 (2024).

**Questions:**

Q1. Is there a particular reason for choosing OpenLM instead of other models such as LLaMa and Qwen adjacent models (those in Table 1)?

---

> ### Author Response · Authors · 2024-11-23
>
> Thank you for recognizing the empirical depth of our work and the practical value of our findings. We're glad you found our extensive experimentation (300+ runs) and ablation studies to be well-designed and sufficient to support our conclusions. Your acknowledgment of the significant improvements (~2% on VQA benchmarks) validates the practical impact of our training guidelines.
>
> # Main Response:
>
> ### Qualitative analysis (Weakness 1):
> > The authors should provide some qualitative analysis, of where the model scoring lower misses compared to other models. Currently, it is not directly clear what kind of questions the 2% drop come from.
>
>
> We apologize for the confusion, this 2% drop from the text baseline was a mistake that will be removed from the paper. The drop in performance from the initial 80% trained model is 10% and from the 100% trained model it is 14%.
>
> To provide further analysis, we detail below the result per benchmark at different stages of our training process. Most text benchmarks see a decreasing score as the model is trained on images and then fine-tuned for visual question answering. Interestingly, AGIEval score actually increases with image training while PubMedQA sees a sharp decrease of the score.
>
>
> | Metric                                      | Ours (4 epochs LLaVA FT) | Ours (No FT) | Base 80% | Base 100% | Llama 3.2 1B | Qwen 2.5 1.5B |
> |--------------------------------------------|--------------------------|--------------|-----------|-----------|--------------|---------------|
> | **AGIEval**                         | 32.5                   | 27.7         | 19.4      | 28.2      | 21.4         | 63.6         |
> | **ARC Easy**                               | 59.3                   | 72.3        | 74.7     | 77.1     | 69.5        | 80.6         |
> | **BigBench Conceptual Combinations**       | 27.2                    | 30.1         | 40.8     | 47.6     | 27.2         | 57.3         |
> | **BigBench CS Algorithms**                 | 35.8                   | 46.2        | 44.6     | 46.7     | 46.5        | 56.5         |
> | **COPA**                                   | 71                   | 83        | 86     | 92     | 83        | 85         |
> | **HellaSwag**                              | 58.2                   | 66.5        | 69.2     | 72.8     | 65.1        | 67.78         |
> | **MathQA**                                 | 22.9                    | 25.9         | 26.9      | 27.3      | 30.52        | 40.84         |
> | **PIQA**                                   | 70.5                   | 74.4        | 76.6     | 79.1     | 76.0        | 76.22         |
> | **PubMedQA**                               | 35.0                    | 38.6         | 66.2     | 69.6     | 65.8        | 66.6         |
> | **Total Stable-Score**                     | 15.44                   | 21.26        | 25.73     | 29.67     | 23.52        | 35.68         |
>
> To help provide a qualitative assessment of our model performance, we also produce some data samples from VQAv2 with different question types https://zenodo.org/records/14201746.

---

> ### Author Response · Authors · 2024-11-23
>
> # Responses (cont.)
>
> ### Additional evaluations (Weaknesses 2 and 3):
> > One concern is that the chosen VQA benchmark is a bit restrictive. The authors should chose a larger set of evaluation benchmark such as MMMU, OCR-Bench to name a few (Qwen2-VL has a good set of possible evaluation benchmarks [Ref1]).
> > Related to previous point, the authors should also include some caption/description as well as OCR heavy datasets as well.
>
> This is a good point – we agree that using only VQA is a bit restrictive.
>
> 1. Our evaluations also include other metrics than general VQA included in our aggregate metric (which we called “stable_score”). We considered a variety of tasks such as TextVQA (document understanding and OCR), RefCOCO (object localization) and OCID-Ref (cluttered object localization). Our evaluation contains at least one benchmark from each of the categories listed in the Qwen2-VL paper (except for video/math/multilingual). See Appendix E for a detailed list of our tasks.
>
> 2. Nonetheless, we agree that using a larger set of benchmarks would be better. To fix this, we add our scores on MMMU below. We are evaluating on other datasets and will add them over the next few days as soon as the results come in.
>
> | Checkpoint | MMMU Accuracy |
> |------------|-----|
> | 90-10 text-image ratio, 100% checkpoint, 4 epochs | 0.3066 |
> | 90-10 text-image ratio, 80% checkpoint, 4 epochs | 0.32 |
> | 90-10 text-image ratio, 60% checkpoint, 4 epochs | 0.327 |
> | 90-10 text-image ratio, 40% checkpoint, 4 epochs | 0.3533 |
> | 90-10 text-image ratio, 20% checkpoint, 4 epochs | 0.24667 |
> | 90-10 text-image ratio, 0% checkpoint, 4 epochs | 0.29333 |
> |PaliGemma-3B-mix-448			          | 0.336 |
> |Qwen2-VL-2B					          | 0.423 |
> |LLaVA-Next-Mistral-7B			          | 0.37 |
>
> The results shown are for our model initialized from the different text pretraining checkpoints, then trained on image text with a mix of 10% image captioning, 90% text, then finetuned for 4 epochs. The compared models, PaliGemma, Qwen-VL have public results for MMMU (https://huggingface.co/spaces/opencompass/open_vlm_leaderboard) we recomputed them to make sure our evaluation pipeline does not make a difference and computed the results in the same way for LLaVA-Next-Mistral. Surprisingly, and as opposed to our previous results, they do not show the 80% checkpoint to be best but it is still stronger than the 100% checkpoint.
>
> ---
>
> # Minor Points:
>
> > The provided methodology is shown on 1B models but not on larger models such as 7B and 70B. While in general it is expected the findings will carry forward, an explicit experiment would be better. However, it might be prohibitively expensive.
>
>
> We are currently running a set of experiments at the 7B scale. However, there’s a chance these experiments may not finish in time for this discussion period. We wanted to get this response out first so that we could begin author-reviewer discussions, but we will share the 7B results as soon as we get them.
>
> > One issue with the guidelines is that these can be only applied at training time by the authors of the original work. Thus, if someone else wants to improve an existing VLM, it requires the original authors to have open-sourced their checkpoints including the optimizers weights and schedule which may be difficult to obtain.
>
>
> We agree that requiring partially trained checkpoints may limit immediate applicability of this paper’s findings. However, we hope this work will encourage open-model contributors to release more than just their final checkpoints in the future. Intermediate checkpoints are likely stored by model developers and could be released if there is sufficient demand, furthering open research. Publishing papers such as ours which shows the value of intermediate checkpoints can help push the field towards this direction.
>
> Moreover, while our findings are directly relevant to labs with access to internal checkpoints, they also provide broader insights into the training dynamics of language models, contributing to the field’s understanding. Importantly, our experiments at smaller scales demonstrate that these insights are applicable even to those working on smaller models, ensuring accessibility beyond frontier labs.
>
> > Is there a particular reason for choosing OpenLM instead of other models such as LLaMa and Qwen adjacent models (those in Table 1)?
>
> This is connected to the point above – we were able to get granted access to the OpenLM (DataComp-LM) checkpoints, whereas the checkpoints for LLaMa and Qwen are not publicly accessible. Also, DataComp-LM was the strongest model of its size when we started this work and is still competitive (stronger than Llama3-1B on many benchmarks).

---

> > ### Comment · Reviewer_Z7KL · 2024-11-24
> > **MMMU experiments suggests 40% instead of 80%?**
> >
> > It looks like the MMMU experiment suggest having mixture of 40% rather than 80% is better.
> >
> > I think this phenomenon needs more exploration, and it isn't immediately clear if the proposed guideline is general enough. Or is it that 80% is better for VQA but not MMMU? Having a comparison of which tasks get better with which model might be quite helpful. For instance, does caption-heavy dataset require some different mix of pre-training?

---

> ### Author Response · Authors · 2024-11-24
>
> Thank you for your follow-up question. The percentages in the MMMU results shown above are not mixtures but the completion of the initial text-only LLM. The rest of the training pipeline remains unchanged: image-text training (mixture of 10% image captioning and 90% text) and LLaVA finetuning.
>
> Here is the list of tasks that show better results with the 80% initial checkpoint: refCOCO, OCID-ref, POPE, textVQA, GQA, VQAv2.
>
> VizWiz shows better results at 100% but with high variance. The model struggles to understand its prompt and often fails to answer appropriately.
>
> MMMU shows better results at 40%. We did not have the time to estimate variance or look at the reason for this surprising result.

---

> ### Author Response · Authors · 2024-12-02
> **Update: 7B results and more holistic evaluations**
>
> Hi, good news! We have some updates to share:
>
> ---
> # 7B experiments
> Our 7B experiments have finally finished, and we are happy to share that our method indeed scales well to the 7B scale. This confirms the main finding of our paper, which is that pre-training with a mixture of image and text data allows models to perform better on downstream tasks.
>
> More specifically, we compared two 7B models as follows:
> 1. Introduce images (at a 90:10 text:image ratio) \~50\% of the way through pre-training for a DCLM-7B checkpoint (2.3T tokens at a 4.3T token schedule). This is done at 1x Chinchilla tokens (~140B tokens). Then fine-tune with Llava for 4 epochs.
> 2. Take the final cooled-down DCLM-7B checkpoint (2.3T tokens cooled-down for 270B more tokens). For a fair token-matched comparison, we continue training with just text at the same token-matched scale as the model above (i.e. 1x Chinchilla tokens). Then fine-tune with Llava for 4 epochs.
>
> Here, model 1 is basically what we're proposing, and model 2 is very similar to how many VLMs are conventionally trained.
>
>
> Our results are as follows:
>
> | Dataset  | Model 1 (with image data in pre-training) | Model 2 (no image data in pre-training) |
> |----------|----------|----------|
> | VQAv2    | **76.39** | 74.69    |
> | GQA      | **61.08** | 60.6     |
> | VizWiz   | 10.88    | **11.69** |
> | TextVQA  | **0.46828** | 0.41624  |
> | RefCOCO  | **0.64168** | 0.58889  |
> | OCID     | **0.46107** | 0.39696  |
> | POPE     | **0.87478** | 0.87113  |
>
> We hope that this gives you a bit more confidence in the validity and scalability of our results. If time permits, we will add even more 7B results to the final version.
>
> ---
>
> # More holistic evaluations
>
> From the suggestions of multiple reviewers, we have also added new evaluations and made it clearer in our paper that we are **not** only evaluating on VQA.
>
> Specifically, aside from the standard VQA tasks, our suite of benchmarks includes:
> - RefCOCO (object localization)
> - OCID-Ref (cluttered object localization)
> - **(NEW)** MMMU
> - **(NEW)** SNLI-VE Visual entailment
>
> With these, we recompute our stable-scores for Section 3.1 and arrive at very similar overall trends:
>
>
> | Checkpoint                                    | Recomputed Stable-Score |
> |-----------------------------------------------|---------------|
> | 90-10 text-image ratio, 100% checkpoint, 4 epochs |     0.3046    |
> | 90-10 text-image ratio, 80% checkpoint, 4 epochs  |   **0.3821**      |
> | 90-10 text-image ratio, 60% checkpoint, 4 epochs  |    0.3426     |
> | 90-10 text-image ratio, 40% checkpoint, 4 epochs  |    0.3644    |
> | 90-10 text-image ratio, 20% checkpoint, 4 epochs  |    0.3349   |
> | 90-10 text-image ratio, 0% checkpoint, 4 epochs   |   0.3269    |
>
> While some individual benchmark-level results may vary (e.g. MMMU), the overall trends remain the same after averaging over the entire suite of benchmarks. More importantly, this once again validates our claim that introducing images into pre-training outperforms the 100\%-checkpoint results.
>
>
> ---
> ..
> ---
> We hope you can reconsider your score in light of these new promising results, and please don't hesitate to let us know if you have any other concerns.

---

### Author Response · Authors · 2024-12-02
**Rebuttal period updates: 7B results and more holistic evaluations**

We thanks the reviewers for all the great feedback. We have integrated these insights, and we feel that these constructive comments really helped strengthen our paper.

We highlight our key changes below:

---
# 7B experiments
We are happy to share that our method indeed scales well to the 7B scale. This confirms the main finding of our paper, which is that pre-training with a mixture of image and text data allows models to perform better on downstream tasks.

More specifically, we compared two 7B models as follows:
1. Introduce images (at a 90:10 text:image ratio) \~50\% of the way through pre-training for a DCLM-7B checkpoint (2.3T tokens at a 4.3T token schedule). This is done at 1x Chinchilla tokens (~140B tokens). Then fine-tune with Llava for 4 epochs.
2. Take the final cooled-down DCLM-7B checkpoint (2.3T tokens cooled-down for 270B more tokens). For a fair token-matched comparison, we continue training with just text at the same token-matched scale as the model above (i.e. 1x Chinchilla tokens). Then fine-tune with Llava for 4 epochs.

Here, model 1 is basically what we're proposing, and model 2 is very similar to how many VLMs are conventionally trained.


Our results are as follows:

| Dataset  | Model 1 (with image data in pre-training) | Model 2 (no image data in pre-training) |
|----------|----------|----------|
| VQAv2    | **76.39** | 74.69    |
| GQA      | **61.08** | 60.6     |
| VizWiz   | 10.88    | **11.69** |
| TextVQA  | **0.46828** | 0.41624  |
| RefCOCO  | **0.64168** | 0.58889  |
| OCID     | **0.46107** | 0.39696  |
| POPE     | **0.87478** | 0.87113  |

We hope that this gives you a bit more confidence in the validity and scalability of our results. If time permits, we will add even more 7B results to the final version.

---

# More holistic evaluations

From the suggestions of multiple reviewers, we have also added new evaluations and made it clearer in our paper that we are **not** only evaluating on VQA.

Specifically, aside from the standard VQA tasks, our suite of benchmarks includes:
- RefCOCO (object localization)
- OCID-Ref (cluttered object localization)
- **(NEW)** MMMU
- **(NEW)** SNLI-VE Visual entailment

With these, we recompute our stable-scores for Section 3.1 and arrive at very similar overall trends:


| Checkpoint                                    | Recomputed Stable-Score |
|-----------------------------------------------|---------------|
| 90-10 text-image ratio, 100% checkpoint, 4 epochs |     0.3046    |
| 90-10 text-image ratio, 80% checkpoint, 4 epochs  |   **0.3821**      |
| 90-10 text-image ratio, 60% checkpoint, 4 epochs  |    0.3426     |
| 90-10 text-image ratio, 40% checkpoint, 4 epochs  |    0.3644    |
| 90-10 text-image ratio, 20% checkpoint, 4 epochs  |    0.3349   |
| 90-10 text-image ratio, 0% checkpoint, 4 epochs   |   0.3269    |

While some individual benchmark-level results may vary (e.g. MMMU), the overall trends remain the same after averaging over the entire suite of benchmarks. More importantly, this once again validates our claim that introducing images into pre-training outperforms the 100\%-checkpoint results.

---

### Meta-Review · Area_Chair_anCf · 2024-12-22

**Metareview:**

This paper conducts a study on recipes for incorporating visual tokens in VLMs. The reviewers like the motivation and relevance, the comprehensiveness of the experiments, and the clarity of presentation. The main concern is on the generalizability of findings given detail on various axes (the choice of benchmark datasets and tasks, the size of the models, etc.). During the rebuttal period, the authors diligently address these concerns, including more holistic evaluation and adding results at the 7B scale, which overall strengthens the paper. The AC believes that the community would benefit from these findings and the discussion that are based upon them, thus recommending acceptance.

**Additional Comments On Reviewer Discussion:**

This paper received scores of 5, 5, 5 and 6. 3 of the 4 reviewers engaged with the authors during the rebuttal but did not raise their scores.

---

### Decision · Program_Chairs · 2025-01-22

Accept (Poster)